# Integration of Lineal Geostatistical Analysis and Computational Intelligence to Evaluate the Batch Grinding Kinetics

Freddy A. Lucay [1], José Delgado [2] and Felipe D. Sepúlveda [2,*]

1   Escuela de Ingeniería Química, Pontificia Universidad Católica de Valparaíso, Valparaíso 2340000, Chile; freddy.lucay@pucv.cl
2   Departamento de Ingeniería en Minas, Universidad de Antofagasta, Antofagasta 1240000, Chile; jose.delgado@uantof.cl
*   Correspondence: felipe.sepulveda@uantof.cl

**Abstract:** The kinetic characterization of the grinding process has always faced a special challenge due to the constant fluctuations of its parameters. The weight percentage of each size (WPES) should be mentioned. There are particular considerations for WPESs, because their tendencies are not monotonic. The objective of this work is to provide a methodology and model that will allow us to better understand the kinetics of grinding through the analysis of the Response Surface (RS), using geostatistical (data reconstruction) and computational intelligence (meta-model) techniques. Six experimental cases were studied and trends were evaluated/adjusted with multiple parameters, including an identity plot adjusted to 0.75–0.90, a standardized error histogram with a mean of $-0.01$ to $-0.05$ and a standard deviation of 0.63–1.2, a standardized error based on an estimated value of $-0.09$ to $-0.02$, a meta-model adjusted to between 92 and 99%, and finally, using the coefficient of variation, which classifies the information (stable/unstable). In conclusion, it was feasible to obtain the results of the WPES from RS, and it was possible to visualize the areas of greatest fluctuation, trend changes, error adjustments, and data scarcity without the need for specific experimental techniques, a coefficient analysis of the fracturing or the use of differential equation systems.

**Keywords:** kinetic grinding metamodel; geostatistics analysis; computational intelligence techniques





## 1. Introduction

In any mining activity, continuous development in the optimization of the process is essential, and this is generally reflected by the acquisition of new equipment, continuous technical updates, and the development of new sensors and algorithms for process control, but in general terms, proposals for new methodologies have been relegated to a subordinate role based on calculations and information analyses. This may be due to multiple factors, but possibly one of the most relevant is the extensive heuristic knowledge that is involved in operational processes.

Currently, the complex conditions arising from mining technical/economic restrictions, environmental regulations, and constant conflicts with nearby communities often mean that the previous knowledge is insufficient; thus, all possible avenues to improve processes are of major interest.

Among the most relevant processes in the area of comminution is the grinding mills, which have the objective of reducing the sizes of the particles to obtain a requested product, but to achieve this, there are varying requirements. These requirements include the following: carrying out the process with the least possible energy consumption (some authors have indicated that between 56% and 70% of energy is used on the comminution of ore, and the energy consumption associated with it is strongly related to the type of ore and the operating factors of the mill), low supply consumption (i.e., the wear of steel balls and steel shells), and low maintenance. These are some of the most important requirements.

However, grinding has a crucial impact on the processes that follow downstream. In the first instance, we found flotation of the minerals, which requires a precise particle size to grant the processes multiple interactions (the relationship between hydrodynamics/physicochemical) and to efficiently promote a selective concentration of the mineral [1–4]. Later, during the thickening, the effect of particle size was on the tailings (sedimentation rate and rheological parameters) and also in the case for the clarified water that was recirculated (including an increase in turbidity due to the presence of colloids, which may cause interference in the flotation process). Finally, the effect was on the tailings dams, where one of the parameters, in both the chemical and physical stability of these deposits [5,6], was the granulometry.

To quantify the particle sizes, multiple techniques are described in the literature (test sieving, laser diffraction, optical microscopy, electron microscopy, elutriation, sedimentation for gravity, sedimentation, and by centrifugal and online particle size analysis [7]), but regardless of which technique is used, the reporting of the data is, in general, carried out by the use of the cumulative weight undersize percentage. This alternative, which is the most widely used due to its ease of calculations to obtain a curve, will represent a defined trend that can obtain a reference size.

With this in mind, its main attribute [7] is related to the sample manipulation protocol. In the case of grinding, it should be considered, if required, to be a delimed, filtered, dried, mineral cake shredding, homogenization process with sample extraction that involves the experimental development of a granulometric distribution analysis. All of the sequences (in addition to the time spent) could generate errors concerning a loss and contamination of the sample when carried out carelessly, and an inadequate samples preparation time is critical (drying and granulometric analysis) and can cause bias in the determination of the granulometric analysis. Secondly, an adjustment to the selected model is important (by linearization of the data). The coefficients of the model are obtained in this way [7], additional to the selected errors that are used to quantify the fit of the model or to quantitatively compare it with other models (the quadratic error is generally used).

However, the reference sizes are very important to quantitatively evaluate the granulometric distribution (GD), and this allows it to be connected to the processes that employ the mass balances, but a reference size in the mass balance may be insufficient, meaning that, in some cases, two or three reference sizes obtain very specific information concerning the granulometric sizes (coarse, intermediate, and fine). However, when it is not possible to obtain good quality information, it is necessary to introduce the concept of population mass balance.

The population mass balance (PMB) can be defined as "population balance or mass size balance", which describes a family of modeling techniques, including tracking and partially or completely manipulating the particle size distribution as they proceed through the comminution process [7]. This is a mathematical technique that is widely used in multiple disciplines [8,9], and it has become increasingly relevant, since it allows us to gain greater knowledge of the comminution process [10–13]. The only drawback lies in its mathematical resolution when trying to obtain its coefficients experimentally (generally using experimental protocols that contemplate mono sizes). They are not easy to obtain, and when looking at the milling process, which is considered to be a chaotic system [7], it becomes even more difficult. This is due to the shocking interactions between particles/balls and walls. This is impossible to predict and could therefore be associated with a significant degree of uncertainty (the system is considered to be nonlinear and nonmonotonic) in the final results. Currently, the population mass balance could be resolved mathematically using dynamic fluid computational techniques [9] that are continually re-evaluated to obtain the most suitable models [7,12,14–16].

GD models exist, generically defined as meta-models [17–19], that adjust an accumulated trend but do not allow for the representation of all the particle sizes. However, there are complex mathematical models that provide this information. This has both experimental and mathematical considerations. However, there may be another alternative, which

has historically not been considered to be feasible due to the complexity of the results of the partial retentions (this is because they are nonlinear and nonmonotonic data) [20]. This alternative is the interpolation technique.

Currently, interpolation techniques have taken the lead due to developments in new meta-models [17–19], the improvement and development of optimization algorithms, and a greater computational capacity, which has allowed the solutions of systems to become increasingly complex.

These points are important at the point the error on a given protocol is analyzed. In general, the occurrences are quantified during the process sampling and its preparation, but there are gaps in how the information obtained is treated, as there is a strong possibility that the process will be smoothed (relaxed) to adjust to the GD meta-models, which is due to the outdated optimization protocols [21].

The present investigation proposes using data from the partial retentions of each size with a geostatistical analysis, as a new way to characterize the grinding mill processes.

To achieve this, it is necessary to carry out a study concerning its spatial trends, which are:

- Reconstruction of the missing information from the response surface.
- Possible meta-models that will represent the response surface with a certainty that it is a global optimum.
- An in-depth analysis of this data and the parameters can be performed.

To carry out all these points, the following mathematical techniques will be used:

- Lineal geostatistical analysis (variograms—Kriging) [22,23] and construction of a meta-model (to support the vector machine and artificial neural networks on a radial basis) [13,24].
- Hybrid mathematical optimization (Particle Swarm Optimization) [25] and descriptive statistics.
- Based on these considerations, this paper will attempt to provide knowledge designed to answer the observations that are available in the literature, such as:
- The grinding process is a chaotic system and exhibits a significant degree of uncertainty, which is difficult to model. However, is it possible to characterize the grinding process using GD and to obtain a trend?
- The PMB requires the determination of several experimental parameters. However, is it possible to propose a different methodology to obtain the weight percentage of each size without using the techniques that were previously described in the literature?

## 2. Methodology

The methodology is composed of four phases of development: Phase I: experimental evaluation, Phase II: Geostatistical evaluation, Phase III: construction of a meta-model and final report, and Phase IV: complementary analysis. These four phases can be seen in Figure 1.

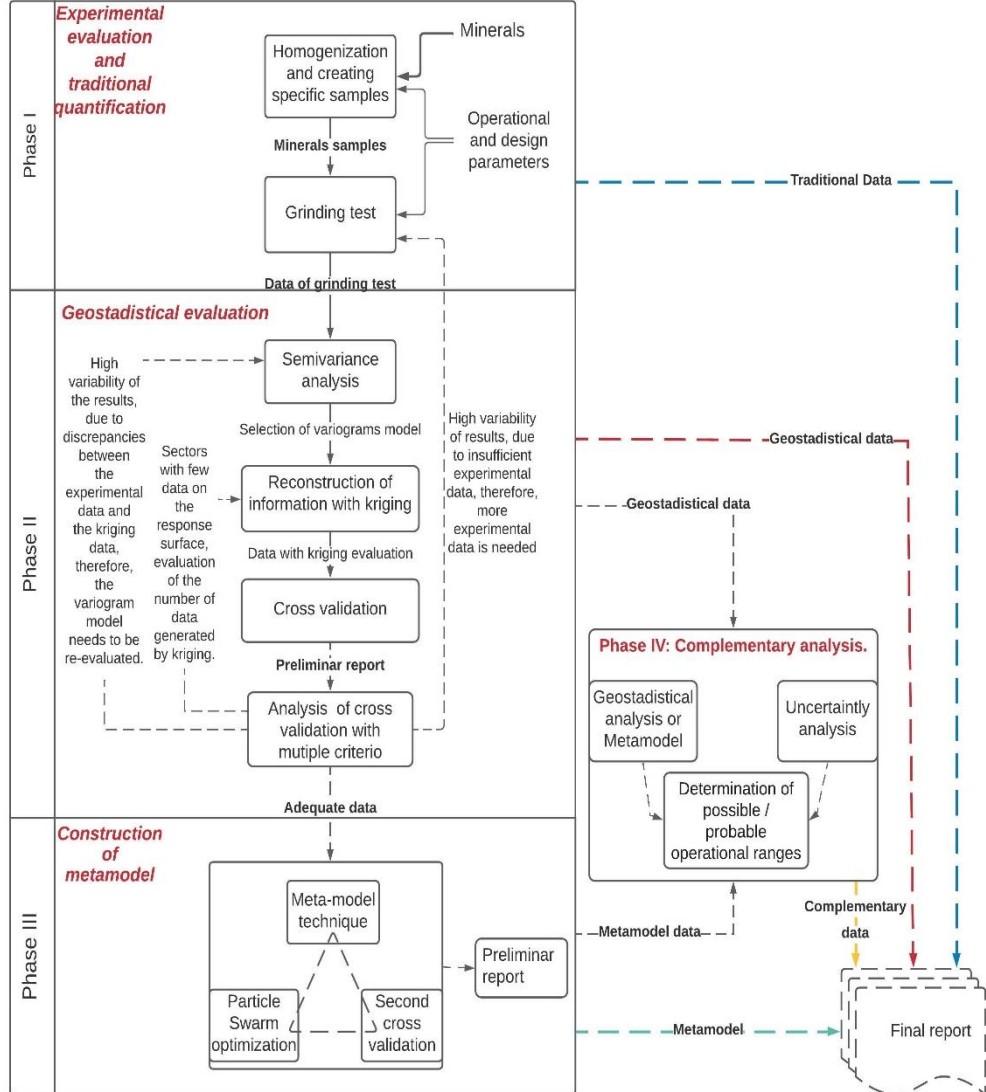

**Figure 1.** Illustration of the methodological proposal.

### 2.1. Phase I: Experimental Evaluation and Traditional Quantification

In the first phase, we considered the treated mineral and the process that was used (taking into consideration the technical and economic capacity to develop it). This characterization can be subdivided into three groups: the first group refers to the mineral. In this group, we found physical assessments (density, hardness, work index, Sag Power Index, porosity, and permeability); chemical assessments (grades of the main and secondary elements and grades of the contaminants); geological assessments (mineralogy, petrography, and lithology); and rock mechanic assessments (fracture index, abrasion index, and rock density, to name a few).

The second group refers to the operation of the process, including the protocols and parameters used, such as the characteristics found in the preparation of the sample (crushing, homogenization, and generation of the samples); the grinding (volume filling, grinding time, ball collar, and mill speed); and product treatment (desliming, drying, and granulometric analysis).

The third group refers to the design of the process, which includes: the type of equipment used (crushers, screens, and mills, to name a few) and its relation to the configuration/sequence of the process. All this information could indicate the possible behavior of the grinding process.

Once the background that characterized the process was obtained, the respective grinding tests were carried out according to their specific protocols. In this way, the necessary information was obtained from the respective masses of each of the meshes, and with this, we calculated what was partially retained and the accumulated intern and modeled the data, in addition to determining the $F_{80}$, $P_{80}$, work index, reduction ratio, and any other required parameter.

After completing all these experimental phases and the protocolized calculations, the second phase of the methodology began.

### 2.2. Phase II: Geostatistical Evaluation

The second phase of the methodology consisted of different mathematical techniques being used to establish a better understanding of the tendency of the comminution. For this, geostatistical techniques were used and analyzed.

### 2.2.1. The Geostatistical Analysis (GA)

The objective of this analysis was to determine the spatial trend of the experimental data. For this, variographic techniques (experimental variogram) were used to obtain the models (modeled variogram) that represented their behavior. Later, reconstruction of the data was carried out using Kriging, and an analysis of the information obtained was performed.

The GA [23,26,27] is a branch of statistics that focuses on the spatial positioning of data. Originally developed to predict the probability distributions of ore grades for mining operations, it is currently applied to multiple areas of engineering and science [28–30] and offers tools to build earth models that are required, for example, by the mining industry, and for the environmental analysis. The main tools include the variogram and different techniques for Kriging.

The Semivariance analysis [26,28] of the Semivariance ($\gamma$) of $Z$ between two data points is an important concept in GA and is defined as:

$$\gamma(x_i, x_0) = \gamma(h) = \frac{1}{2} var[Z(x_i) - Z(x_0)] \tag{1}$$

where $h$ is the distance between points $x_i$ and $x_0$, and $\gamma(h)$ is the semivariogram (commonly referred to as variograms). The variograms (Vr) provide information about the size of the search window used in the spatial interpolation methods. In other words, it provides the structural analysis for the spatial interpolation. Therefore, Vr can be estimated from the data as follows:

$$\hat{\gamma}(h) = \frac{1}{2n} \sum_{i=1}^{n} (Z(X_i) - Z(X_i + h))^2, \tag{2}$$

where $n$ is the number of pairs of the sample points that are separated by distance $h$. The Vr models may consist of simple models, including Nugget, Exponential, Spherical, Gaussian, Linear, and Power models, or the nested sum of one or more simple models [26,31] or a complex model [32].

Reconstruction of the information with ordinary Kriging: This section of the methodology seeks to use the concept of spatial prediction (this concept includes interpolation and extrapolation analysis) or spatial interpolation, which is defined as the "area the studies use to create surface data based on a set of sampled points" [33].

Spatial interpolation techniques are varied and have multiple combinations, but in general, they can be divided into three categories: (1) non-geostatistical methods, (2) geostatistical methods, and (3) combined methods [28].

For this methodology, we focused on the methods included in the second category. For this, we used what is known as the best linear estimation ($x$) of an unbiased area ($V$). In this case, the Kriging [18,28] was defined as linear, because its estimations were calculated through linear combinations that were weighed against existing data in Equations (3) and (4). It was also unbiased, because it tried to neutralize the error average.

The general aim of Kriging is to interpolate data in a continuous space. The main advantage of Kriging is that it can provide better estimates on data minerals than its counterparts, which are considered to be traditional methods (arithmetic mean, polygon, and inverse distance weighting), therefore, its main fundamentals are: (1) The traditional methods use Euclidian concepts, where the distance is the main consideration, but Kriging considers the distance, as well as the geometry from where the samples are taken. (2) Using Kriging minimizes the error variance (difference between the real value and the estimated value), and in this way, it prevents systematic errors during calculation. The geostatistic methods are very flexible in the sense that data can be extrapolated and useful when estimating point values or blocks. It can also include complementary information that is related to the main variables, which can be expressed as:

$$Z_k(x) = \sum_{i=1}^{N} \lambda_i z(x_i), \tag{3}$$

Inside the area under scrutiny, $V$, Kriging attributes the weight ($\lambda_i$) to each of the material grades (in this case, the grades are sizes), which are obtained from samples $z(x_i)$ inside coordinate $x_i$ and, in this way, obtains an estimate of the material size $Z_k$ in zone $V$. The weights $\lambda_i$ are calculated with the sole purpose of minimizing the error variance, where $\lambda_1 + \lambda_2 + \cdots + \lambda_N = 1$; therefore, a high weight value is defined for close samplings, while lower values are used for samplings that are further away in zone $V$. Therefore, to obtain the Kriging equations, we have to minimize the expression of $\sigma_E^2$.

$$\sigma_E^2 = 2\sum_{i=1}^{N} \lambda_i \frac{1}{V} \int_V \gamma(x_i, x)dx - \frac{1}{V^2} \int_V \int_V \gamma(x, y)dxdy - \sum_{i=1}^{N} \sum_{j=1}^{N} \lambda_i \lambda_j \gamma(x_i, x_j), \tag{4}$$

The $i$th the restriction:

$$E\lfloor Z_K(x) - z(x_i) \rfloor = 0, \tag{5}$$

Finally, because it uses different linear equations and resolves the equations in the system, Kriging can use different methods in its application, which include ordinary Kriging and simple Kriging. The first uses the local average from the points being sampled and a position variable. The second uses the average from all the existing points inside the evaluated zone. However, while these two types of Kriging are widely used, other types of Kriging can also be used when ordinary and simple Kriging cannot obtain reliable results.

### 2.2.2. Cross-Validation

This stage refers to the use of multiple techniques that are designed to obtain the smallest error variance, both locally and globally. As an example, techniques, such as the comparison of identity graphs, use histograms and descriptive statistics [34].

### 2.2.3. Geostatistical Analysis Report

Finally, the information reported in the variographic and Kriging analysis can be complemented with cross-validation, where the correlation of the experimental data and that generated by Kriging can be established. This provides a confidence interval in the data. Another analysis that we can include is the spatial uncertainty analysis, which, among other things, uses the standard deviation of the spatial data throughout the analysis and detects the sectors with the greatest uncertainty.

### 2.3. Phase III: Construction of Meta-Model and Final Report

The main objective of the last phase is to improve the information by using mathematical assembly techniques. In this case, it is a meta-model assembly (this is better known as surrogate assembly), and its objectives are [35–37]: (1) to identify regions where we expect large uncertainties or errors (contrast), (2) to provide a more robust approach, (3) to use an ensemble of surrogates via the weighted average (combination) or the selection of the best surrogate model based on error statistics, which would provide a better approximation than

individual surrogates, and (4) to gain a better understanding of the relationship between x and y and the extent of its application [38–40].

Therefore, the assembly in this work is focused on the use of geostatistic data, the generation of superficial models, and the application of heuristic optimization techniques.

Meta-Model Technique

The meta-modeling techniques [41,42] are techniques that are essentially related to data gathering and can generate approximated models. The most commonly used meta-models include: the polynomial model, splines model, general linear model, general additive model, Kriging model, Gaussian process meta-modeling, neural network model, and regression tree model.

The Least Squares Support Vector Machine (LS-SVM) is used for regression problems [43]. According to the literature, the support vector machine used for the regression problems is defined as: given a training dataset of $N$ points $\{x_k, y_k\}_{k=1}^N$, where $x_k \in \mathbb{R}^n$ are the input data and $y_k \in \mathbb{R}$ are the corresponding prediction values; we can construct a nonlinear function ($f$) in the form:

$$y_k = f(x_k) = \langle w, \varphi(x_k) \rangle + b, \tag{6}$$

where $\varphi(\cdot)$ is a nonlinear function that maps the input space into a higher-dimensional space, $w \in \mathbb{R}^n$ and $b \in \mathbb{R}$. It is difficult to ascertain how the equalities and errors $e_k$ in Equation (7) are introduced:

$$\langle w, \varphi(x_k) \rangle + b = y_k - e_k, \ k = 1, \dots, N, \tag{7}$$

The unknowns, $w$ and $b$, can be determined via the following problem:

$$\min_{w,b,e} J(w,b,e) = \frac{1}{2} w^T w + \frac{1}{2} \gamma \sum_{k=1}^N e_k{}^2, \tag{8}$$

Subject to the equality constraints:

$$\left(w^T \cdot \varphi(x_k) + b\right) = y_k - e_k, \ k = 1, 2, \dots, N, \tag{9}$$

where $\gamma$ is a regularization factor. The Lagrangian function of the optimization problem earlier is:

$$\mathcal{L}(w,b,e;\alpha) = J(w,b,e) - \sum_{k=1}^N \alpha_k \cdot \left(\left(w^T \cdot \varphi(x_k) + b\right) - y_k + e_k\right), \tag{10}$$

where $\alpha_k$ are the Lagrange multipliers. Using the Karush–Kuhn–Tucker conditions, it is possible to determine $\alpha_k$ and $b$ via solving the matrix system. However, this depends on the $\gamma$ and kernel used. We utilized the kernel $K(x, y) = \phi \cdot \exp\left(-\alpha \cdot \|x - y\|^2\right) + (1 - \phi) \cdot tanh(\beta \langle x, y \rangle + r)$, where $\beta > 0$, $\alpha > 0$, $r < 0$, $0 < \phi < 1$ and $\langle \varphi(x), \varphi(y) \rangle = K(x, y)$ [44,45]. The factors $\gamma$, $\beta$, $a$, $r$, and $\phi$ were determined using Swarm Intelligence.

Swarm Intelligence (SI) [46]: The SI is included in the field of artificial intelligence. This consists of a population of simple agents that interact locally with one another and with their environment [47]. In this approach, inspiration often comes from nature, especially biological systems [48]. Well-known examples of SI include ant colonies, animal herding, bacterial growth, fish schooling, and bird flocking [47]. This latter is the basis of the particle swarm optimization (PSO).

PSO was proposed by Kennedy and Eberhart in 1995 [46] and considers an objective function ($f$) that must be minimized. The goal is to find a solution $p^{best}$ (particle) for which $f(p^{best}) \leq f(x)$ for all $x$ (particles) in the search space, and this could mean $p^{best}$ is the global minimum. Here, each particle´s movement is influenced by its best local position and the best positions in the search space. The particle´s movement is updated as better

positions are found by other particles. The position of the particles in PSO is updated according to its velocity:

$$v^i_{j+1} = w \cdot v^i_j + c_1 r_1 \cdot \left( p^{best}_j - x^i_j \right) + c_2 r_2 \cdot \left( p^i_j - x^i_j \right), \quad (11)$$

$$x^i_{j+1} = x^i_j + v^i_{j+1}, \quad (12)$$

where $v^i_j$ denotes the velocity of the *i*th particle in the *j*th iteration of the search procedure, $x^i_j$ denotes the position of the *i*th particle in the *j*th iteration of the search procedure, $p^i_j$ is the best position of the *i*th particle in the *j*th iteration of the search procedure, $p^{best}_j$ is the best position of the swarm in the *j*th iteration of the search procedure, $w$ is the inertia weight used to balance the global exploration and local exploration, coefficients $c_1$ and $c_2$ are positive constant parameters called acceleration coefficients, and $r_1$ and $r_2$ are uniformly distributed random variables within a certain range [0,1].

### 2.4. Phase IV: Complementary Analysis

Phase IV has the objective of integrating/comparing the knowledge acquired from the different types of mathematical analysis (geostatistical analysis, statistical analysis, meta-model construction, and uncertainty analysis). In this way, it is possible to integrate the different observations/conclusions/recommendations to obtain an integral knowledge of the process under study.

For this phase, we will evaluate the results utilizing the coefficient of variation. (*Cv*), the common parameter to evaluate the uniformity on mixing, which is defined as:

$$Cv = \frac{\sigma}{\overline{X}}, \quad (13)$$

where $\sigma$ is the standard deviation and $X$ is the average. The smaller the *Cv*, the more uniform the mixture is. However, it presents different categorizations according to the different applications that have been used [49–54].

Now, Koch & Link 1971 [54] mentioned three important characteristics where the use of *Cv* is relevant:

(1) "The number of ore samples required to obtain a specified precision of estimate for an unsampled ore deposit". However, in our case, it would be the number of samples to obtain the best spatial tendency of the comminutions.

(2) "The coefficient of variation is a guide to the form of the statistical distribution that should be applied for data analysis" Within the proposed methodology, can be applied this comment as an additional calculation to confirm an observed trend.

(3) "The coefficient of variation is a useful guide for quality control of sampling". This last comment may be particularly relevant when there is no prior information to determine the number and time lapses required for sampling.

### 2.5. Final Report

This provides detailed information on the results from the grinding milling process, and this comes from the multiple mathematical analyzes carried out, both in the parametric and graphical forms, and may be used to make decisions or to connect them with other mathematical techniques that are outside this methodology.

### 3. Cases of Studies

The case studies analyzed below are of two minerals with different characteristics. They come from two mining companies that are located in the second region of Chile.

*3.1. Fase I: Experimental Evaluation and Traditional Quantification*

Minerals: The input information for each ore is described below: first, ore-A: 182.5 kg of the material, $F_{80}$, 1388 μm, a specific gravity of 2.84, and a W.I. of 14.9 kWh/short ton. Second, ore-B: 276.5 kg of the material, $F_{80}$, 875 μm, a specific gravity of 2.78, and a W.I. of 11.6 kWh/short ton. Third, ore-A is a strata-bound copper deposit and is hardly altered. Meanwhile, ore-B is a porphyry copper deposit that is highly altered.

A geological analysis of the samples of ore-A was developed by a Motic SMZ-171 electronic magnifying glass; an eyepiece 10X(Ø23)/magnification with a range of 0.75X–5X, which detects the presence of quartz, plagioclase, orthoclase, biotite, and muscovite, representing 98% of the samples; and the remaining 2% consists of chalcosine, bornite, molybdenite, chalcopyrite, and pyrite.

A geological analysis of the samples of ore-B was developed by a Qemscan and detects the mineralogical elements, including chalcopyrite, bornite, chalcanthite/digenite, pyrite, the chlorite group, biotite/phlogopite, and the kaolinite group.

Operational and design parameters: The general parameters of the laboratory grinding protocol (LGP) of three mining companies located in Chile are: LGP-1: 138 iron balls that have 1″ diameters, with a total load weight of 10.22 kg, a roller at 70 rpm, and a solid percent of 67%. LGP-2: 145 iron balls are of different sizes, including 1 1/2″, 1,″ and 7/8″, with a total load of 9.34 kg, a roller at 70 rpm, and a solid percentage of 50%. LGP-3: 238 iron balls are of different sizes, including 1″, 7/8″, 3/4″, and 1/2″, with a total ball load weight of 6.66 kg, a roller at 70 rpm, and a solid percentage of 67%.

Homogenizing equipment: The mechanical homogenization equipment used includes a Rotary table splitter DR-10 of Labtech-HEBRO, which allowed for division in six containers, with a maximum capacity of 6 kg. The second piece of homogenization equipment used was a pro-splitter, which allowed for the division of 30 containers with a capacity of 300 g. Both types of equipment have discharge hopper vibrations and receptacle movement with variable speeds.

Grinding equipment: The equipment used was a standardized ball mill, a laboratory scale with a capacity of 5.2 L, and a roller-HEBRO (variable RPM).

Desliming equipment, a drying oven, and granulometric analysis: Desliming equipment was used to remove fine particles by wet sieving with a 200 Ty mesh, the Labtech-Hebro brand machine, and its products were dried at 95 °C for 12 h. The granulometric analysis was carried out in a Ro-tap W.S. Tyler, model RX-29-10, the meshes used were the following: #10, #20, #30, #40, #50, #60, #70, #100, #140, #200, #270, #325, and #-325, which includes all the Tyler series.

The homogenization and production of specific samples: The homogenization process was the same for ore-A and ore-B, and they were based on the laboratory protocol for the mining industry to homogenize and create the samples that were evaluated in the mills and flotation. Therefore, samples of 940 g were generated from ore-A, and samples of 1404 g were generated from ore-B.

Multiple cycles were developed to obtain both materials in a completely homogenized form, and the effect of the variability of comminution was only due to the grinding process itself.

Grinding test and GA: The grinding process began by checking the size and weight of the steel balls, and later, considering the load of the respective balls, the mineral, and finally, the water was charged to the driving roller. The HEBRO was activated and checked that 70 rpm was reached and that the timer control was adjusted.

In the case of the three-dimensional analysis, grinding tests were performed based on a three-minute sampling of the different types of LGP, and a response surface concerning their partial retentions was obtained.

In the case of the two-dimensional analysis, ten repetitions of the grinding test were performed as a function of the specific times to be evaluated; thus, the duration of the samples were 3, 9, and 12 min for the combination of ore-A/LGP-1.

When the equipment completed the different grinding operations, the content of the material was discharged into a receptacle that had holes in it to separate the steel balls from the pulp; the obtained pulp was delimed (200 # Ty), and the final product was dried for up to 12 h at 95°.

All the experimental tests of GA were carried out on the previously declared mesh sizes, and all the materials obtained from the grinding process were analyzed.

A preliminary report is included in this phase, in addition to the existing data, which incorporates the general calculations of the GA.

Preliminary report: As a first preliminary report, it was found that, when using the same experimental protocol and considering the input data of ore-A and ore-B, the evolution in the comminution process was different, and it provided an evolution of a much finer material for ore-B than for ore-A (experimental data in Tables A1 and A2). This is shown in Figure 2.

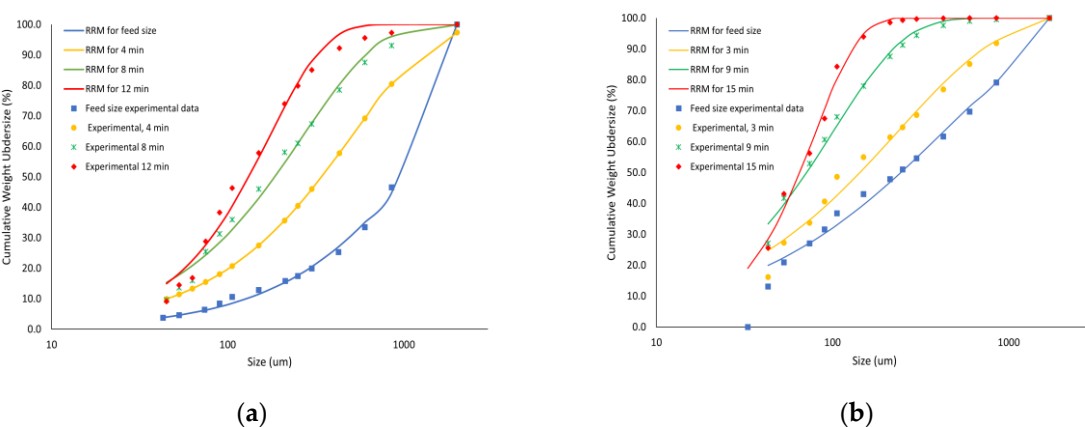

**(a)**                                                                          **(b)**

**Figure 2.** Semi-log plot of GD for feed minerals and when it was used LGP-1 for (**a**) ore-A and (**b**) ore-B.

For the granulometric adjustment analysis, the two most common types of models were evaluated: the Rosin-Rammler Model (RRM) and the Gate-Gaudin-Schuhmann Model (GGSM). Both were adjusted by the least-squares minimization and constitute the RRM model, which is the one that presents the smallest error. Through this, it was possible to determine the respective $P_{80}$ and the optimal grinding times for the flotation tests. This was the case for both ore-A and ore-B, as shown in Table A3.

### 3.2. Phase II: Mathematical Evaluation

Different mathematical programs, but the most relevant was the MATLAB, Isati [47], Gslib [55], Sgems [48], and Excel programs, which were used on a computer with Inter (R) Core (TM) i5-7200K 2.7 GHz and 8 GB of RAM.

The input data is positioned in a mesh generated by the residence time, which is located on the x-axis, and the opening sizes of the sieves are located on the y-axis (two-dimensional position in Annex 1). The results from the retained particles are shown on the z-axis. It is important to remark that the y-axis was taken in millimeters, because micrometers are not accepted as distance units in the Isati, Gslib and Sgems programs.

General analysis: This part concerns some considerations in the calculation development of the 1 and 2-dimensional systems. The first element that can be mentioned is the position of the data in space, which shows an irregular spacing (mesh) within the plane (Figure 3a) and has a greater proximity to the information in the area of small sizes (x-axis). This is due to the opening system of the ASTM mesh (geometric progression) [20]. Another relevant point is the dimensioning of the parameters on the x-axis, as well as the y-axis. For this particular case, the x-axis (time) shows units and mathematical values that are larger

than those shown on the y-axis (mm). In this way, scale problems between both structures are generated.

To solve the second problem, the dimensions of the shafts must first be adjusted with the coefficient (constant of 18 in the y-axis) that was incorporated (Figure 3b). In this way, it was possible to make the dimensions of the axes compatible (arithmetic progression), in addition to the information generated becoming acceptable to the Gslib geostatistical program [55]. Another method could be used that normalizes the axes [56]. To solve the first problem, this methodology was carried out.

Other relevant aspects include the input data and the deepening of statistical information, as can be seen in Figure 4, where:

- The general parameters, such as the mean, variance, and the upper quartile, do not show a significant difference between the two cases. This could obviously generate appreciation errors due to insufficient analysis and could lead to hasty conclusions being made.
- It is only when partially accumulated histograms are observed that a more noticeable difference could be seen. This is the case when there is a greater accumulation of information on small sample sizes (mesh 140# Ty to −325# Ty).
- It is important to remark that these two results cannot be extrapolated to other cases, as shown in Figures A1 and A2. In each particular case, the basic statistical data and their respective histograms differ considerably; therefore, this is evidence of the effect of the implications of the ball collar chosen [57–59].

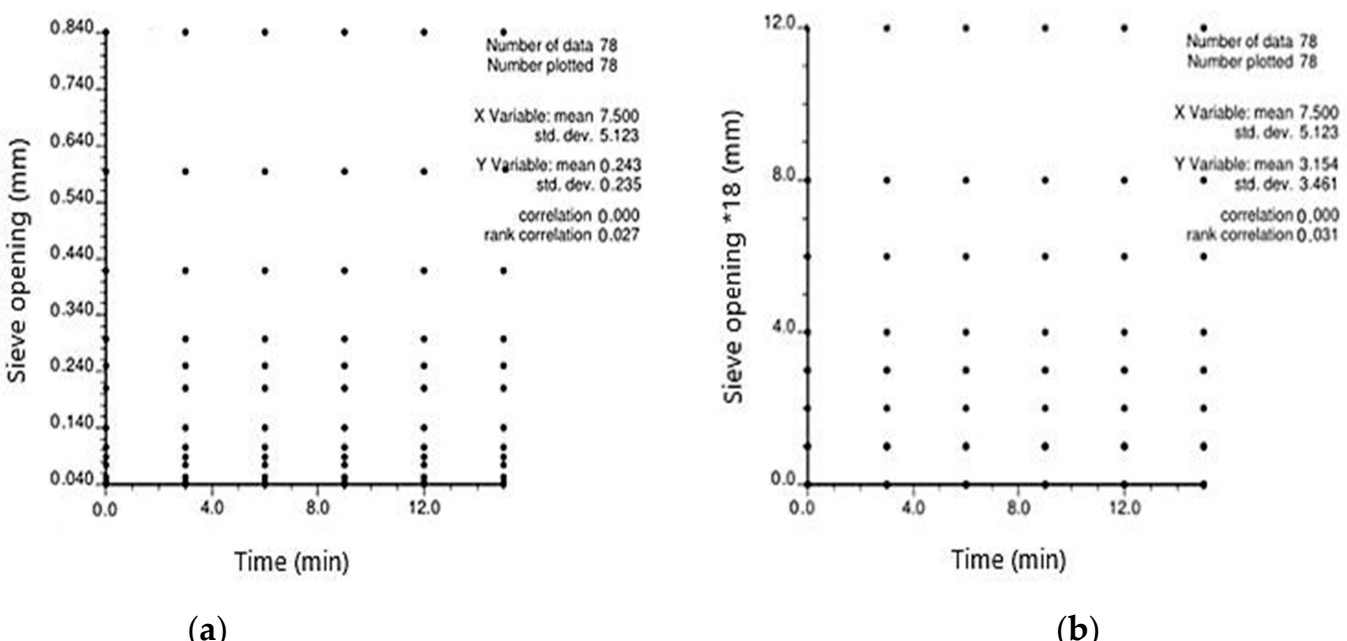

**Figure 3.** Spatial positioning of the experimental data: (**a**) No adjustment of the spatial position (on the Y-axis, it is presented in this format by default for the GSlib program) and (**b**) adjusting the spatial position.

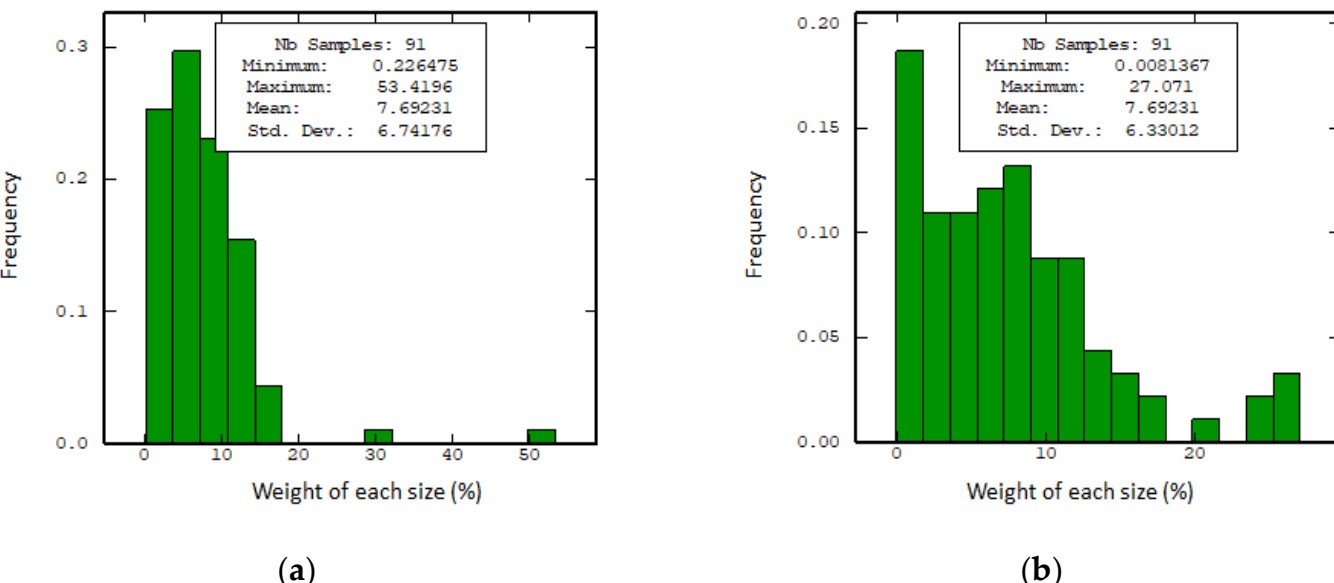

**Figure 4.** Histogram of the input data using LGP-1: (**a**) Ore-A and (**b**) ore-B.

Currently, it is well-known that the information is concentrated in small sizes, but the question to be asked is: How will the LGP affect the grinding kinetics? In this case, it is not possible to determine this using the previous statistical analysis. Therefore, it is necessary to quantify it from a spatial (three-dimensional) point of view.

Three-Dimensional Analysis

Semivariance analysis: This analysis was carried out in two ways: As a first comment, it was the result of an omnidirectional analysis [49] and was later carried using the multi-directional analysis (using different directions for your analysis), which had a poor result in the first analysis of the retrofit, and for the multidirectional analysis, better options for selecting a variogram model were detected. Figure 5 shows this difference.

Further, the experimental variograms obtained allowed us to observe a clear anisotropy, and these presented significant differences. For example, in Figure 5a, a drift component is present, while, in Figure 5b, it has a greater influence on the geometric anisotropy. These two trends were seen in all the data analyzed and were differentiated by the type of ore evaluated.

According to the variographic analysis, the models that best represent their spatial tendencies are called the "composite models" [28,32], and the parameters of the variogram model for both directions (Figure 5) were: an angular tolerance of 45°, a lag of 2.5, a count of 10, and a tolerance lag of 50%. Additionally, the model for ore-A (Figure 5a) was a nonstationary model that was composed of three structures: the nugget effect has a sill: 3, a lineal scale with (5, 5, 12) and gaussian scale (15, 25, sill 20). The model of ore-B (Figure 5b) showed anisotropy, which can be interpreted as geometric—that is, it had different ranges but had a similar plateau and variability within the analyzed area. The model used had two structures: a nugget effect has a sill: 15, a spherical structure has a sill: 35, and in addition to a direction scale (5, 20) (the data in parentheses, correspond to parameters specific to each of the variographic models [23,32], being the traditional way of describing them).

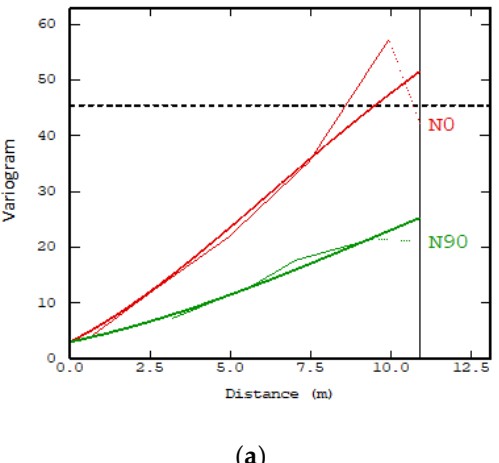

(**a**)

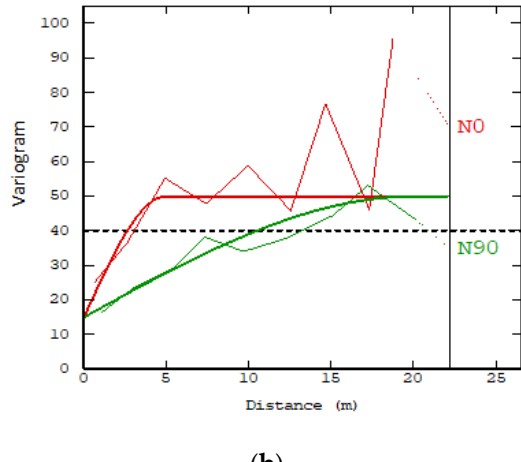

(**b**)

**Figure 5.** Experimental (thin line) and modeled (thick line) anisotropic variograms: (**a**) LGP-1 for ore-A with directions: north: 0° (red) and directions: north: 90° (green); (**b**) LGP-1 for ore-B with directions: north: 0° (red) and directions: north: 90° (green).

Reconstruction of information: Through the variographic analysis, it was possible to apply the ordinary Kriging in order to reconstruct the missing data in the response surface. For this, an 80% confidence was considered; its spatial tendencies are presented in Figure 6a for LGP-1 and Figure 6b for LGP-2. In this case, the differences between the spatial trends were clearly presented, and it showed a much smoother and clearer spatial trend on the right direction of the information in the case of LGP-1, unlike the data obtained for LGP-2.

Geostatistical analysis: In this part of the analysis, we focused on two main subjects: first, the quality of the reconstruction of the missing data within the studied area, and the second was to detect any new information that could be incorporated in the report.

For the first analysis, it can be mentioned that the data generated by the variogram/Kriging analysis was recalculated several times. This iterative process was due to the fact that there was no single variogram that fitted the existing information. A model was chosen to fulfill four criteria in the cross-validation: scatter plot (identity line) between real and modeled values, standardized error histogram, standardized error based on the estimated value histogram, and statistical parameters of the estimated data. All these criteria must be compatible in order to provide the best estimate of the data so as not to produce a bias or manipulation of the spatial trend of the data.

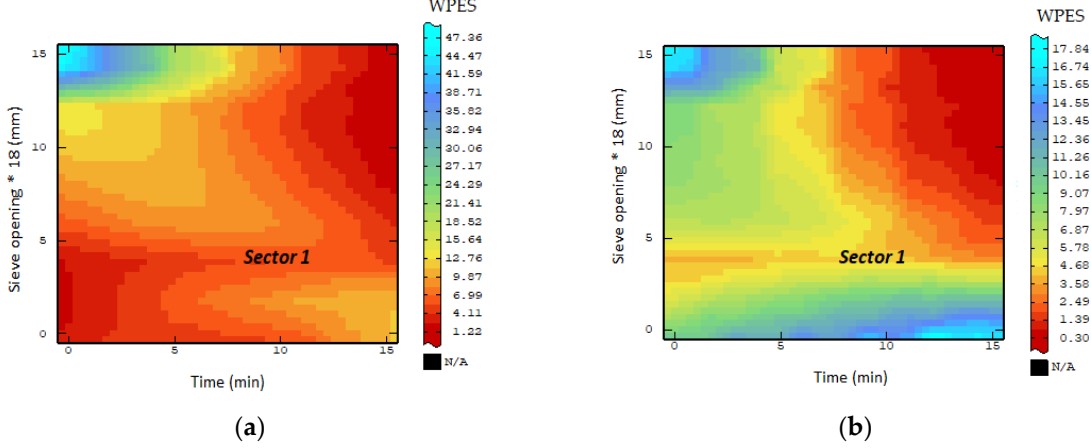

(**a**)　　　　　　　　　　　　　　　　　　　(**b**)

**Figure 6.** Spatial interpolation through a geostatistical analysis: (**a**) LGP-1 for ore-A; (**b**) LGP-1 for ore-B.

In ore-A (Figure 7a), the dispersion of the data is mainly focused on what was retained from 5% to 15%. The data show scattered points, but all these are within the trend. Compared to ore-B (Figure 8a), where up to 10% was retained, ore-A had a good fit, but some data that were presented later were outside this trend.

In regard to the standardized error, it is shown that ore-A (Figure 7b) has a mesokurtic trend, and there are some data outside the confidence level at both ends of the histogram; however, the highest concentration of data is mainly located in the left sector of the histogram. For ore-B (Figure 8b), the data has a leptokurtic trend, and data points outside the confidence level are presented in the left section of the histogram.

For the standardized error based on the estimated data in ore-A (Figure 7c), there are data outside the confidence level, both in the upper and lower limits. However, this is shown in the entire study range for the lower limit, while, in the upper limit, they are in the zone with the highest percentage of what was partially retained. For ore-B (Figure 8c), the data outside the confidence level were at the lower limit, and a high percentage was partially retained.

Finally, the histogram and its statistical data can be compared qualitatively/quantitatively to the histograms of the original data (the last analysis was performed to avoid errors such as biases in the data generated). For ore-A (Figure 7d) and ore-B (Figure 8d), a good similarity is observed in their respective histograms, as well as in their statistical parameters. Therefore, an observer could think that he/she could be in the presence of one of the best variograms (it can be a local optimum), but there is no certainty that it is the global optimum.

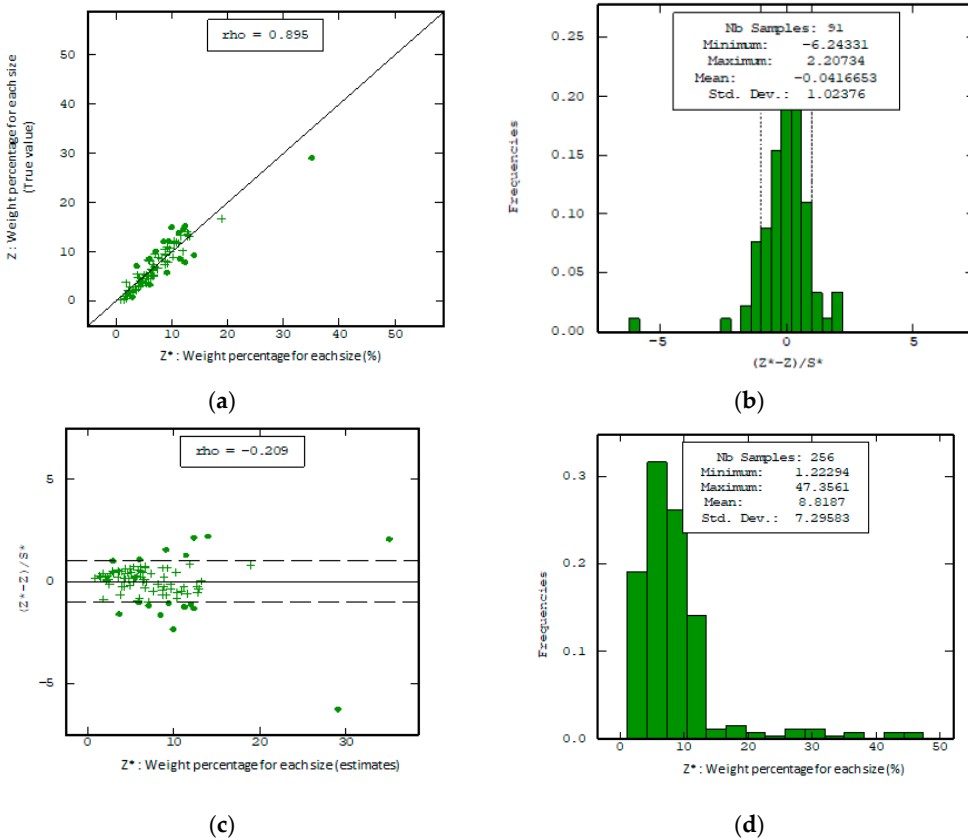

**Figure 7.** Cross-evaluation of the geostatistical analysis of LGP-1 in ore-A: (**a**) the scatter plot (identity line) on the x-axis is the modeled data, and the y-axis shows the experimental data; (**b**) standardized error histogram; (**c**) standardized error based on the estimated value; and (**d**) histogram and the statistical parameters of the estimated data.

In these cases, although a good correlation was determined (in addition to Figures A5 and A7), this does not mean that they have similar performances in all cases, as can be observed in Figures A6 and A8, where their adjustments were very poor and were the product of several possible causes, including experimental data that were detected well outside the trend. In this case, it is recommended that this test be repeated. Another element is the impossibility of finding a suitable variogram. Finally, the experimental protocol could have generated a greater uncertainty in some specific grinding tests.

As a second analysis, the results of the spatial standard deviation are presented (Figure 9), where their respective fluctuations are observed, showing that, for ore-A, its fluctuation is slightly less than the fluctuation of ore-B; this could be a possible criterium to incorporate other grinding times, which were not contemplated in the original program. In this way, the quality of the geostatistical adjustment and analysis could be improved.

Another comment that could be made on Figure 9 is that it clearly shows that the density of the information directly affects the quality of the standard deviation of the Kriging. The areas with higher densities of information have lower deviations, as compared to the less dense areas, but this is strongly related to the inhomogeneous scale [7,20] used for the quantification of the particle sizes.

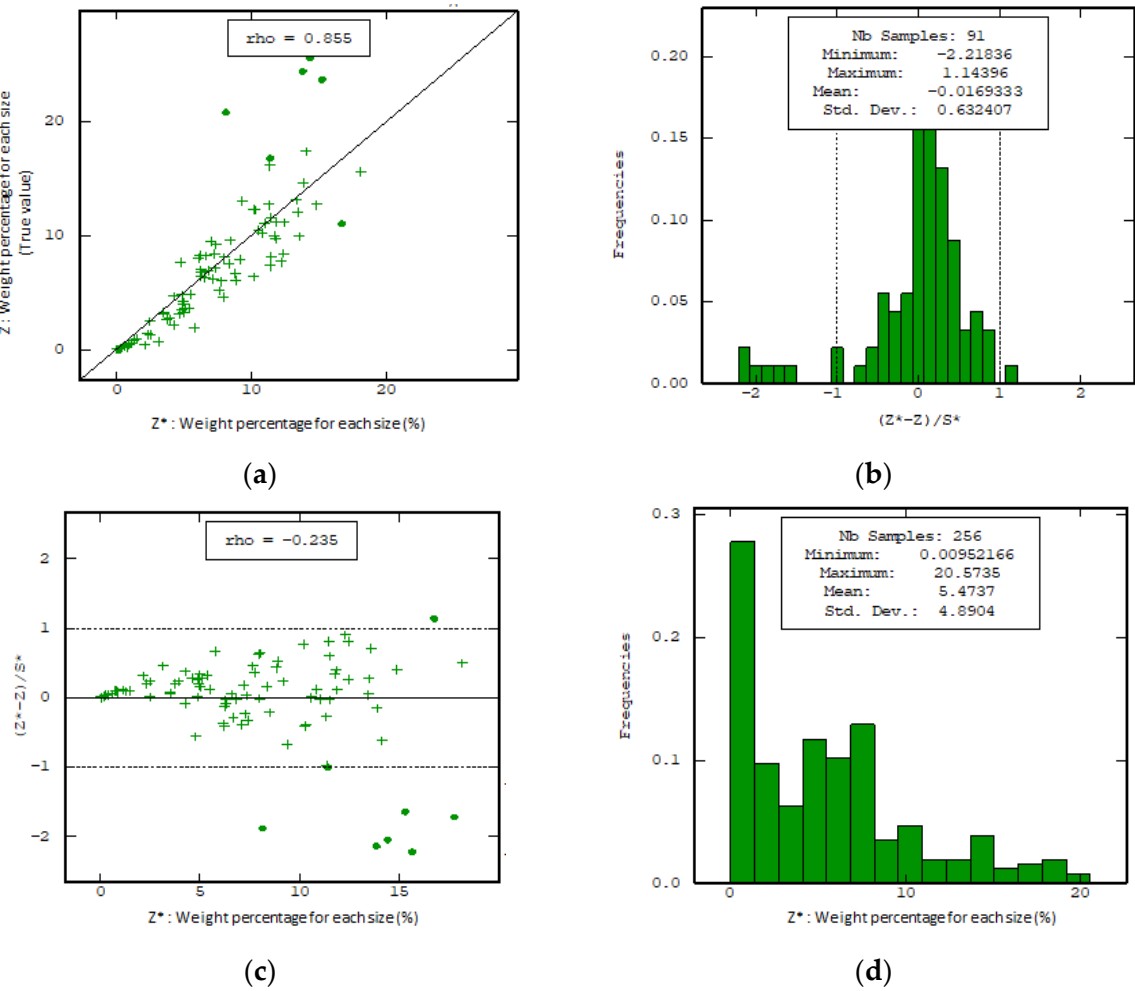

**Figure 8.** Cross-evaluation of the geostatistical analysis of LGP-1 with ore-B: (**a**) scatter plot (identity line) on the x-axis is the modeled data, and the y-axis shows the experimental data; (**b**) standardized error histogram; (**c**) standardized error based on the estimated value; and (**d**) histogram and its statistical parameters of the estimated data.

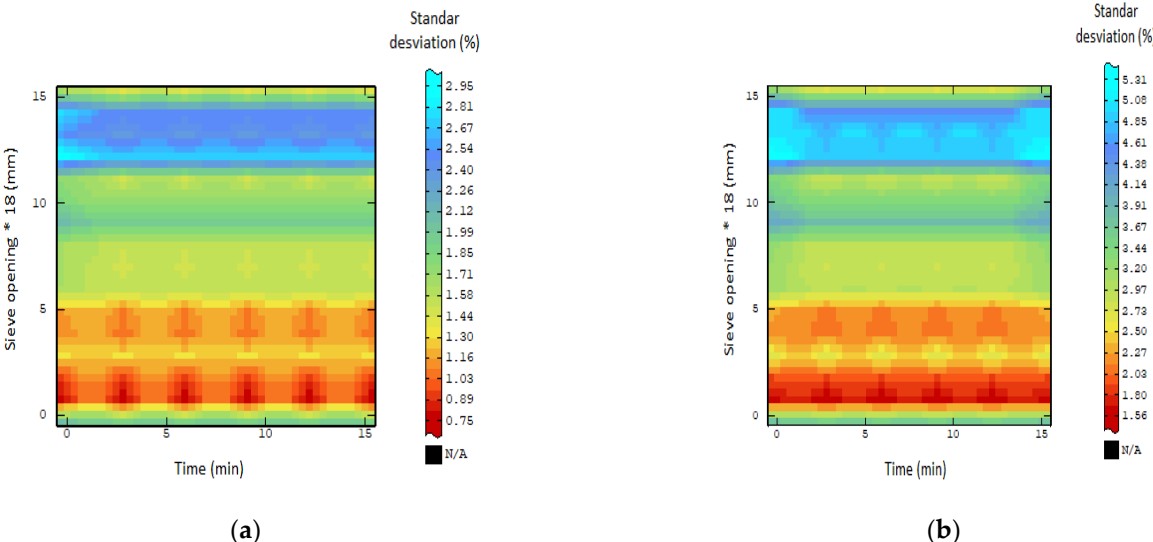

(a) (b)

**Figure 9.** Spatial evolution of the standard deviation of the reconstructed data: (**a**) LGP-1 for ore-A; (**b**) LGP-2 for ore-B.

### 3.3. Fase III: Construction of a Meta-Model and Final Report

For this phase, the most relevant program was R-software on two computers; the first computer was used to perform the algorithm and for supervised training of SVM (exploratory analysis) using a computer with Intel Core i7 2.21 GHz and 16 GB of RAM. The second computer was used for calculating the cases reported in the publication (determination of meta-model) using a computer with a CPU core I9-9900K 5GHz, RAM Corsair DDR4 16 GB 3000 MHz Vengeance, and, finally, the first computer was used to generate the final graphs (cases studio).

The objective of this phase is to pass from discontinuous data to a meta-model that represents the entire response surface. For this, datasets obtained from the geostatistical analysis (256 data for both minerals) were used to train the LS-SVM via PSO. Subsequently, the experimental datasets (91 data for both mineral) were used to test LS-SVM. The first thing that must be reported is the mesh that was used for the adjustment of the model. As shown in Figure 10, the blue points are the coordinates that were used to adjust the meta-model, and the red points are the experimental data; the geostatistical data were generated with a displacement in normalized time with a normalized size of 0.05. Therefore, this is not reported, because there is a saturation in the graph.

The LS-SVM parameters obtained from the adjustment were $C = 142.180$, $\alpha = 0.001$, $\beta = 14.43$, $r = -452$, and $\phi = 0.298$ for ore-A. For ore-B, the LS-SVM parameters were $C = 1000$, $\alpha = 0.001$, $\beta = 0$, $r = 0$, and $\phi = 1$. Additionally, for ore-A, the following results were obtained: a three-dimensional diagram (Figure 11a), a contour diagram (Figure 11b), an identity diagram between the meta-model and the geostatistical data (Figure 11c), and an identity diagram of the geostatistical data and the experimental data (Figure 11d).

The first thing to highlight is the fit that was obtained between the meta-model and the geostatistical data/experimental data, which had an adjustment of over 92%; therefore, the model could provide a good approximation and the trend obtained could be representative of the milling process. Additionally, through the use of the meta-model, it is possible to obtain a better graphical evaluation of the trend, for example, in Figures 7a and 12b.

Per the analysis shown in Figure 11a, it is possible to observe much more of the non-linear and monotonic trend in all the grinding time evaluations, and it is also possible to define three zones: the first zone is from the initial time to 5 min, where it has a more stable tendency. The second zone that can be noticed is a transition zone from 5 min to 10 min (high fluctuation), and finally, the last zone also has a more stable trend, but a high amount of material above 850 μm was observed due to the presence of the peak within the trend.

The result obtained for ore-B (Figure 12) shows a better fit than ore-A, which was over 94.24% (Figure 12c) and had some points outside the trend in the middle region of what was partially retained. However, it was possible to observe a great fluctuation in this mineral, as compared to ore-A. Finally, the trends in Figures 12, A6 and A8 confirm the great complexity of the results that could be obtained from the altered (soft) minerals during the grinding process, as compared to the less-altered minerals that are independent of the LGP used.

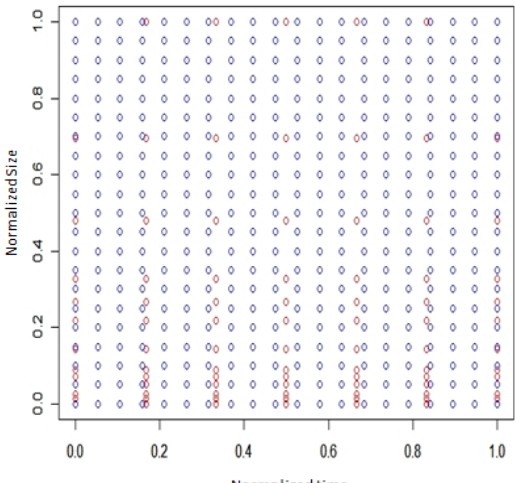

**Figure 10.** Spatial positioning of the experimental data (red) and meta-model data (blue).

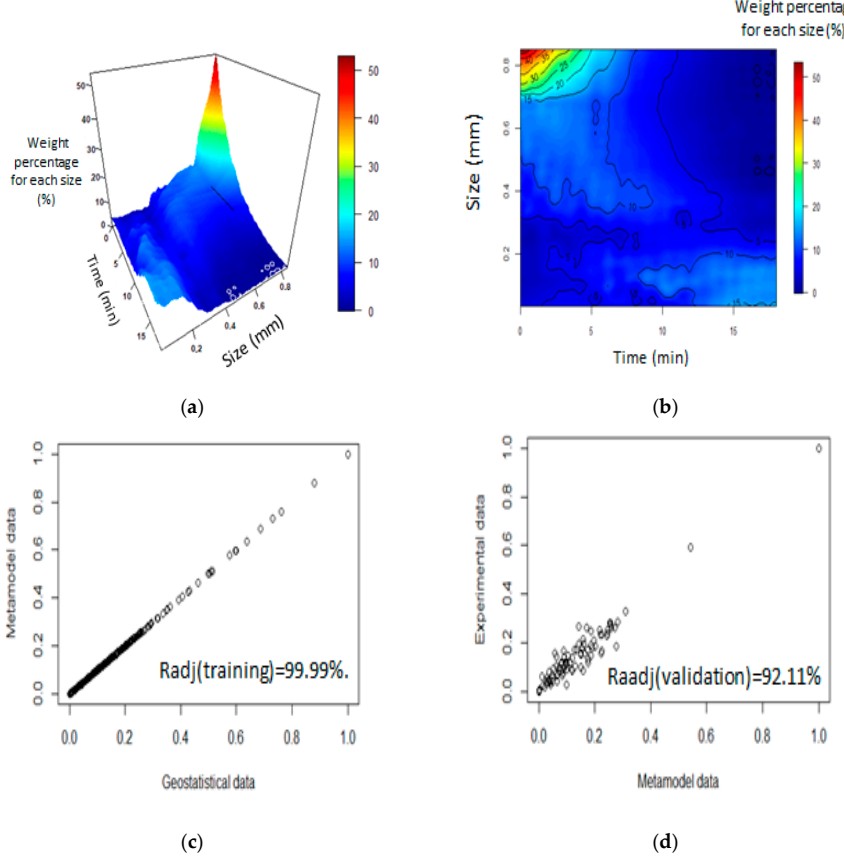

**Figure 11.** Results obtained for LGP-1 for ore-A. (**a**) Spatial trend with normalized data. (**b**) Contours graphic. (**c**) Cross-analysis with the meta-model data and geostatistical data. (**d**) Cross-analysis with the experimental data and meta-model data.

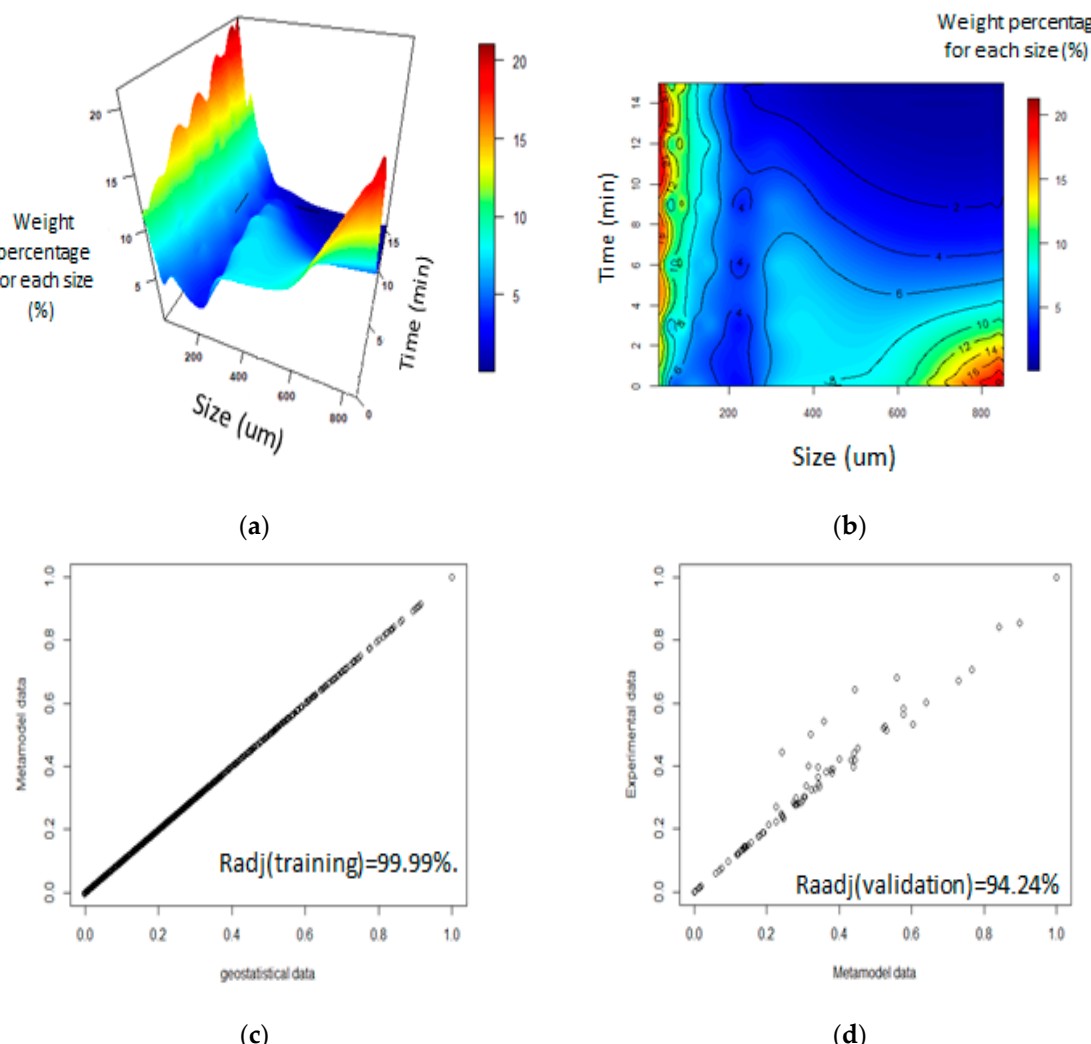

**Figure 12.** Results obtained for LGP-1 for ore-B. (**a**) Spatial trend with normalized data. (**b**) Contours graphic. (**c**) Cross-analysis with the meta-model data and geostatistical data. (**d**) Cross-analysis with experimental data and meta-model data.

*3.4. Phase IV: Complementary Analysis*

In this phase, the $Cv$ was evaluated in three types of analysis, the first analysis is the evolution of $Cv$ concerning the data obtained from the geostatistical estimation (categorizing the limits based on mining economics [50]), the second is related to the data obtained from an uncertainty analysis of LGP-1 ore-A [60], where it was evaluated for each size and at specific times, and the last evaluation is the composite of the data associated with the grinding times.

For the first analysis, the exploratory analysis of the data is performed, showing the differences in the dispersion of the data between the LGP-1 ore-A and ore-B (Figures 13a and 14a); it is possible to clearly observe a zone that presents a reduction of the dispersion; in the case of ore-A, this is approximately at 9 min, while, in ore-B, it is presented at 4 min. This effect is present in all the cases studied (Figures A9a,c and A10a,c) having similar dispersion reduction times but with different tendencies, this being associated to the different grinding surfaces obtained (the dotted lines being a range established at 95% confidence).

Now, the dispersion analysis is much clearer with the $Cv$, where, for ore-A, Figure 13b presents several classification zones of the data fluctuations, being presented in all the cases studied (Figures A9b and A10b). While, in the case of ore-B (Figure 14b), its data can be classified as chaotic, being the same when analyzing the case of LGP-2 (Figure A9d) but

in the case of LGP-3 (Figure A10d). This information can be considered relevant when evaluating certain desired milling conditions ($P_{80}$), since we could be working in a zone of high variability, and the general criteria of operation control may not be applicable (in addition to considering the propagation of the uncertainty of each control parameter).

For the second analysis, the $Cv$ was analyzed for each of the sizes and as a function of time (Figure 15, above), and it can again be observed that there are fewer points with a lower $Cv$ value for the 8 min grinding time compared to the values obtained in 4 and 12 min of grinding; therefore, this parameter can also be manifested as an additive contribution of uncertainties [60,61]. In addition, particular sizes are observed that present a higher value of $Cv$, which may also be a consideration when defining a specific size for process control.

Finally, when analyzing the composited $Cv$ data (Figure 15, bottom), the same trend as in Figure 13b is obtained, but considering the general statistical criterion [49], showing again that there is this zone of lower fluctuation of the information; therefore, independent of the technique with which the input data are obtained for analysis (geostatistical or uncertainty experimental analysis), it is possible to detect potential zones where the process achieves greater stability to the detriment of the other zones.

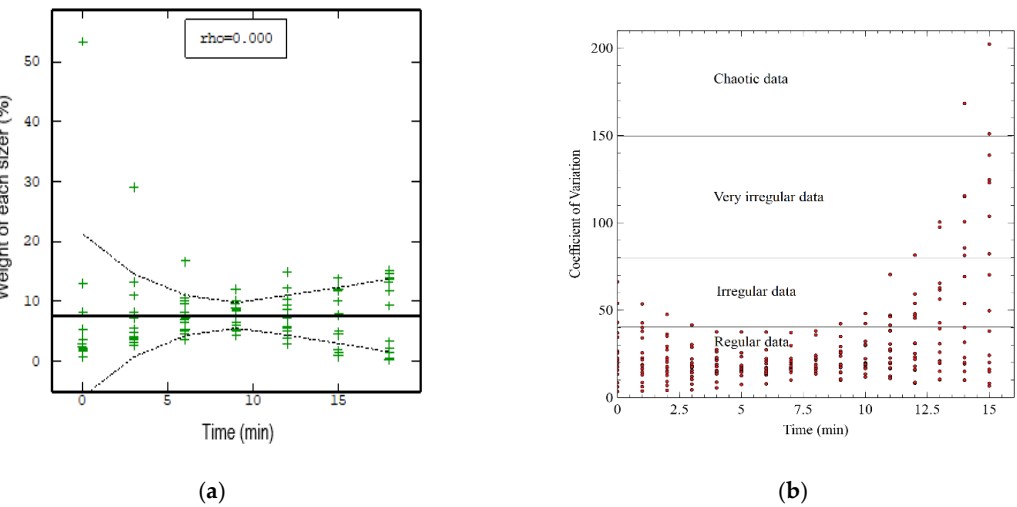

(a)                    (b)

**Figure 13.** Results for LGP-1 for ore-A for time evolutions: (**a**) weight of each sizer fluctuation zone of the reconstructed data; (**b**) $Cv$ for each time.

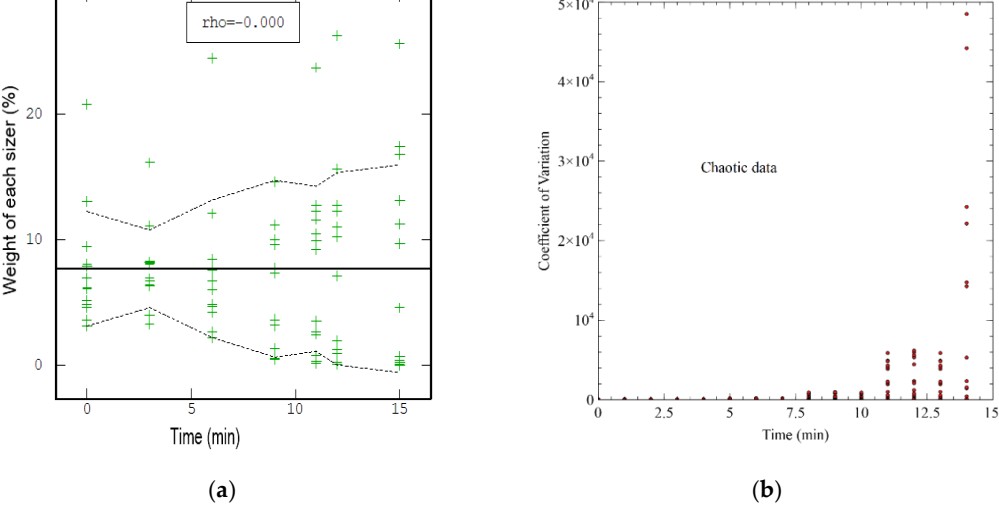

(a)                    (b)

**Figure 14.** Results for LGP-1 for ore-B for time evolutions: (**a**) weight of each sizer fluctuation zone of the reconstructed data; (**b**) $Cv$ for each time.

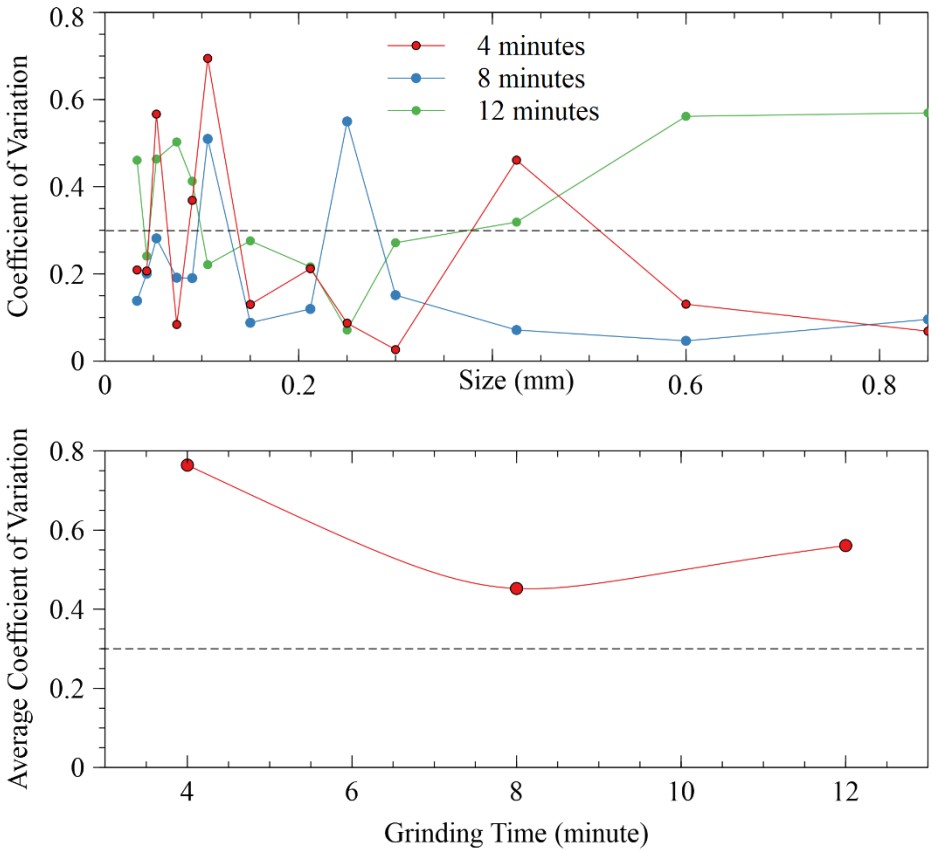

**Figure 15.** Coefficient of variation for LGP-1 for ore-A: (**above**) weight of each sizer, and (**bottom**) average of *Cv* for each grinding time.

*3.5. Final Report*

As presented in the methodology descriptor, the final report has all the information obtained from all the analyses obtained, being, in this case, the contents of all the different analyses, observations, and conclusions that were reported.

**4. Conclusions**

In general terms, it was possible to generate a methodological proposal that could provide an alternative to the current analysis of GD in order to obtain WPES. It could, through a meta-model, also be an alternative for PMB that is associated with a geostatistical analysis and computational intelligence techniques.

In terms of spatial analysis, the complexity of the minerals that produce more alternance than the others were shown to generally be softer/broken, and at the same time, these minerals present highly fluctuating trends in the area under investigation (ore-B) (this may be due to the modification of the physical mechanism of comminution or could predominately be the result of impact/compression/abrasion or a mixture of them). In some cases, they generate areas with more than one change in the trend, and this might explain the high rate of variability that has been reported for these types of materials [14,20]. Despite the previous consideration, the kinetic evolution of the milling process has a defined spatial trend (shown in the variographic analysis), and this depends on the characteristics of the minerals used, in addition to the work protocols used and the variability of the process itself. This could be improved by increasing the number of experimental tests that are conducted in relation to the maximum uncertainty of the quantifiable points (uncertainty analysis for each specific grinding time) or by using the spatial standard deviation (Figure 10) obtained from the geostatistical analysis.

An important consideration that was noted in this paper is that the geostatistical analysis necessarily requires an interactive/cyclical evaluation due to the multiple criteria that can be considered in the cross-evaluation, and this is strongly dependent on the experience of the engineer or researcher who analyzes the data. This observation is directly related to the multiple considerations that are relevant when using geostatistical programs (Sgems, Isati, etc.), since they are not designed for the analysis of this type of information. However, its use allows for significant improvements to be made to the analysis of the data. The analysis technique managed to adjust to different ranges, including the identity plot, which adjusted to the range of 0.75–0.90, the standardized error histogram, with mean of −0.01 to −0.05 and a standard deviation of 0.63–1.2, a standardized error based on the estimated value of −0.09–0.02, and the statistical parameters of the estimated v/s experimental data. The adjustments are variable, since they depend on the confidence limits that are self-imposed in order to develop the calculations. Finally, in the analysis of the estimated v/s experimental histograms, it is visually more difficult to compare, but its direct comparison uses scatter plots (identity adjustment).

The construction of a meta-model that represents the grinding kinetics could be developed, but it would require, in addition to experimental data, being paused at different stages of grinding, as well as additional generated data through geostatistics [56].

The use of normalized axes or adjustment factors is required in order to develop the linear geostatistical analysis; this is because the concept of the "nominal opening size" of the mesh follows the geometric progression. However, in the case of linear geostatistical techniques, the maximum use of the different geostatistic/statistic techniques is obtained through an arithmetic progression; in this way, the data are distributed in an "approximately" uniform space, thereby reducing the concentration of the information in specific sectors.

Another point to consider is the number of data necessary; using excess data could generate an overload in the work area, thus producing overfitting of the meta-model and consequently resulting in a very slow calculation that is not necessarily better. Additionally, this could result in greater errors in the adjustment for the same numbers of data.

An important consideration concerning the quality of the adjustment is that the grinding process continues to have an uncertainty associated with its z-axis. If a statistical study of its variability of this z-axis is not generated as a function of specific times (uncertainty analysis), its error may be within the intrinsic variability of the process.

Furthermore, obtaining the meta-model turns out to be a combination of mathematical techniques that are compatible, and they can have a good fit between the experimental and modeled data, with an adjustment of over 92% (considering that the surface under study had a complex tendency) and an adjustment of 99.999% in the geostatistical analysis; therefore, the analyses of each of the parameters associated with cross-evaluation (scatter plot, standardized error histogram, standardized error based on the estimated value, and histogram and its statistical parameters of the estimated data) can be considered to be compatible. Therefore, in this way, it was possible to determine, at a particular grinding time, a series of sizes or one size of interest that was urgently needed. In addition, it is important to remark that, when a meta-model is available, it can be used in multiple analyses and calculations. The main disadvantage is the computation time. For example, the geostatic data process can be generated in a few seconds, while the determination of the meta-model can easily take several hours or days.

Finally, the use of parameters such as *Cv* to evaluate from another point of view the variability of the information allows us to have a global idea of the potential fluctuations, its different zones, the range of its variability in the milling process, a recommendation of the number of samples being analyzed, and its tendencies confirmed between the correlation of the geostatistical data and a previously published research of experimental uncertainty analysis, but if it should be evaluated, which are the adequate ranges of how the information is classified, since, in this publication, they were evaluated through techniques that perhaps were not completely compatible with the grinding process.

**Author Contributions:** Conceptualization, F.D.S.; methodology, F.D.S.; software, F.D.S., F.A.L. and J.D.; validation, F.D.S., F.A.L. and J.D.; writing—original draft preparation, F.D.S.; writing—review and editing, F.D.S., F.A.L. and J.D.; funding acquisition, F.D.S.; and project administration, F.D.S. All authors have read and agreed to the published version of the manuscript.

**Funding:** The authors thank the support of ANID through the Fondecyt program, grant no. 11180328.

**Institutional Review Board Statement:** Not applicable.

**Informed Consent Statement:** Not applicable.

**Data Availability Statement:** Data is contained within the article.

**Acknowledgments:** Special thanks to Pamela Godoy Plaza, Mauricio Garcia Morales and Dubett Muñoz Calderon for their experimental support.

**Conflicts of Interest:** The authors declare no conflict of interest.

## Appendix A

**Table A1.** Experimental data for the weight of each sizer (%) for LGP-1 ore-A.

| | Grinding Time (min) | | | | | | |
|---|---|---|---|---|---|---|---|
| Size (μm) | 0 | 3 | 6 | 9 | 12 | 15 | 18 |
| 850 | 53.14 | 28.99 | 16.66 | 8.89 | 3.86 | 1.33 | 0.26 |
| 600 | 13.08 | 13.24 | 10.10 | 6.15 | 2.91 | 0.92 | 0.23 |
| 425 | 8.16 | 11.00 | 10.69 | 8.51 | 5.09 | 1.92 | 0.46 |
| 300 | 5.42 | 8.19 | 9.57 | 10.10 | 8.58 | 4.67 | 1.60 |
| 250 | 2.44 | 3.62 | 4.51 | 5.36 | 5.67 | 4.50 | 2.20 |
| 212 | 1.68 | 2.75 | 3.61 | 4.45 | 5.11 | 4.98 | 3.31 |
| 150 | 2.93 | 5.45 | 8.28 | 8.78 | 10.43 | 12.14 | 11.79 |
| 106 | 2.26 | 4.84 | 6.99 | 7.76 | 8.59 | 11.96 | 14.61 |
| 90 | 2.16 | 3.95 | 7.36 | 12.06 | 15.00 | 10.14 | 13.98 |
| 74 | 2.06 | 3.07 | 6.51 | 9.65 | 12.26 | 11.78 | 13.26 |
| 53 | 1.80 | 4.06 | 5.21 | 6.59 | 9.39 | 7.86 | 9.33 |
| 43 | 0.83 | 3.61 | 5.35 | 6.48 | 7.33 | 13.90 | 13.69 |
| 33 | 3.76 | 7.13 | 5.08 | 5.16 | 5.79 | 13.89 | 15.25 |

**Table A2.** Experimental data for the weight of each sizer (%) for LGP-1 ore-B.

| | Grinding Time (min) | | | | | |
|---|---|---|---|---|---|---|
| Size (μm) | 0 | 3 | 6 | 9 | 12 | 15 |
| 850 | 20.80 | 8.12 | 2.18 | 0.45 | 0.07 | 0.01 |
| 600 | 9.48 | 6.75 | 2.68 | 0.57 | 0.10 | 0.02 |
| 425 | 8.09 | 8.21 | 4.84 | 1.35 | 0.27 | 0.05 |
| 300 | 6.99 | 8.27 | 7.69 | 3.24 | 0.96 | 0.24 |
| 250 | 3.62 | 4.00 | 4.71 | 3.18 | 1.28 | 0.41 |
| 212 | 3.14 | 3.25 | 4.20 | 3.64 | 1.95 | 0.72 |

**Table A2.** *Cont.*

| Size (μm) | Grinding Time (min) | | | | | |
|---|---|---|---|---|---|---|
| | **0** | **3** | **6** | **9** | **12** | **15** |
| 150 | 4.86 | 6.44 | 8.44 | 9.60 | 7.14 | 4.59 |
| 106 | 6.21 | 6.32 | 7.57 | 9.99 | 12.73 | 9.70 |
| 90 | 5.18 | 8.02 | 6.01 | 7.37 | 12.25 | 16.81 |
| 74 | 4.59 | 6.93 | 6.71 | 7.73 | 10.25 | 11.26 |
| 53 | 6.06 | 6.39 | 8.44 | 11.18 | 11.05 | 13.14 |
| 43 | 7.90 | 11.14 | 12.09 | 14.64 | 15.65 | 17.45 |
| 33 | 13.06 | 16.17 | 24.45 | 27.07 | 26.29 | 25.61 |

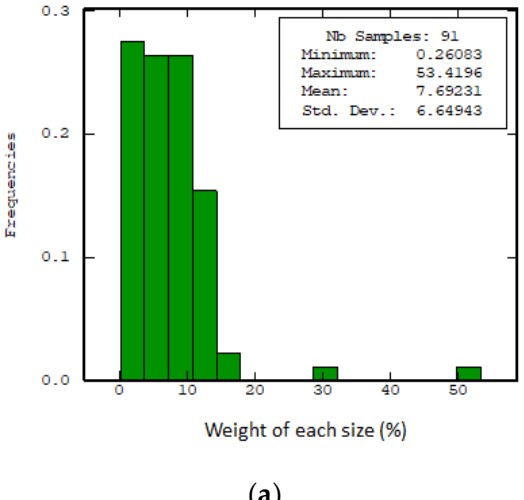

(**a**)

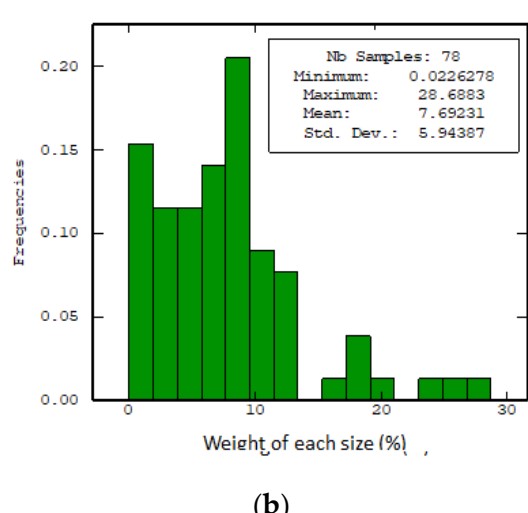

(**b**)

**Figure A1.** Histogram of the input data using LGP-2. (**a**) Ore-A. (**b**) Ore-B.

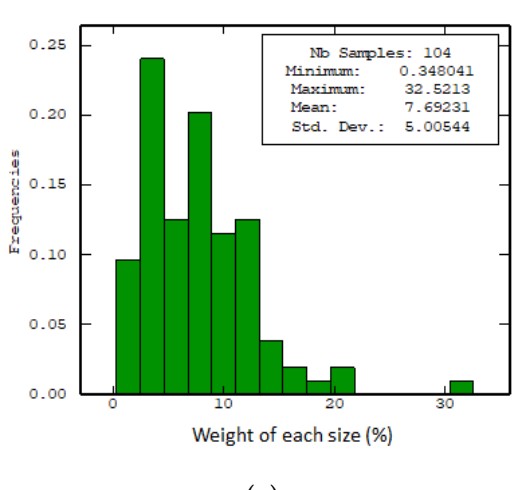

(**a**)

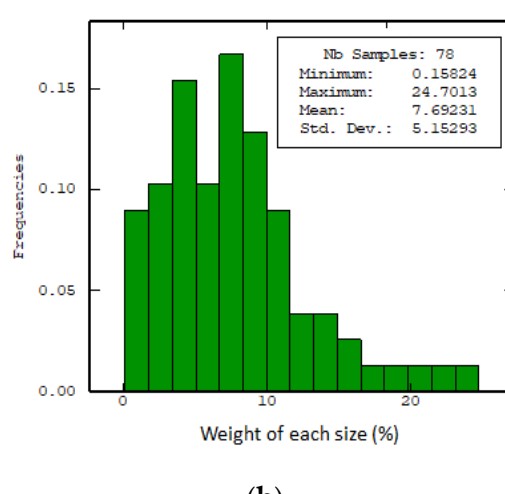

(**b**)

**Figure A2.** Histogram of the input data using LGP-3. (**a**) Ore-A. (**b**) Ore-B.

**Table A3.** Results of P80 with RRM.

| Time (min) | P80 (μm) | |
| --- | --- | --- |
| | ore-A | ore-B |
| 3 | | 486 |
| 4 | 864 | |
| 8 | 463 | |
| 9 | | 162 |
| 12 | 234 | |
| 15 | | 102 |

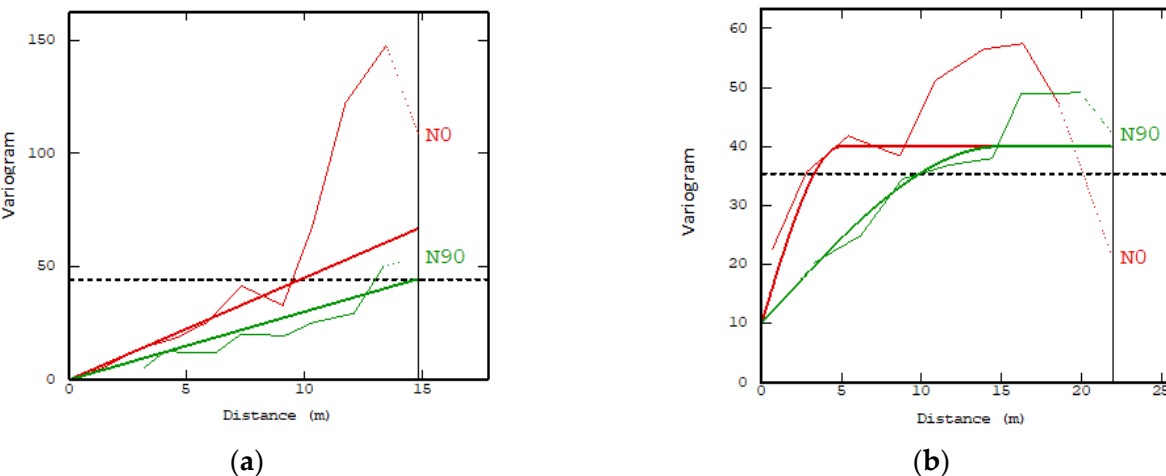

(**a**)　　　　　　　　　　　(**b**)

**Figure A3.** Experimental (thin line) and modeled (thick line) data with anisotropic variograms: (**a**) LGP-2 for ore-A with directions: north: 0° (red) and directions: north: 90° (green); (**b**) LGP-2 for ore-B with directions: north: 0° (red) and directions: north: 90° (green).

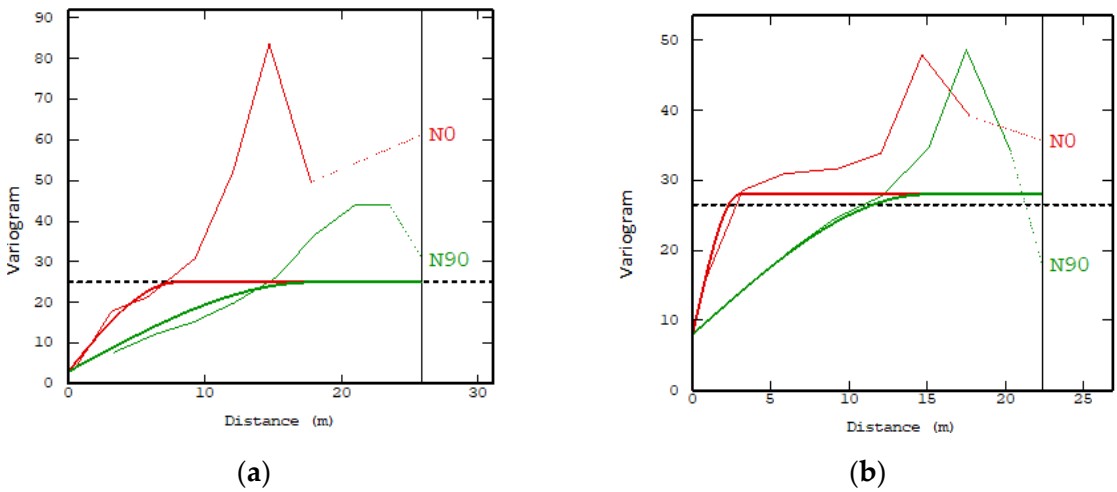

(**a**)　　　　　　　　　　　(**b**)

**Figure A4.** Experimental (thin line) and modeled (thick line) data with anisotropic variograms: (**a**) LGP-3 for ore-A with directions: north: 0° (red) and directions: north: 90° (green); (**b**) LGP-3 for ore-B with directions: north: 0° (red) and directions: north: 90° (green).

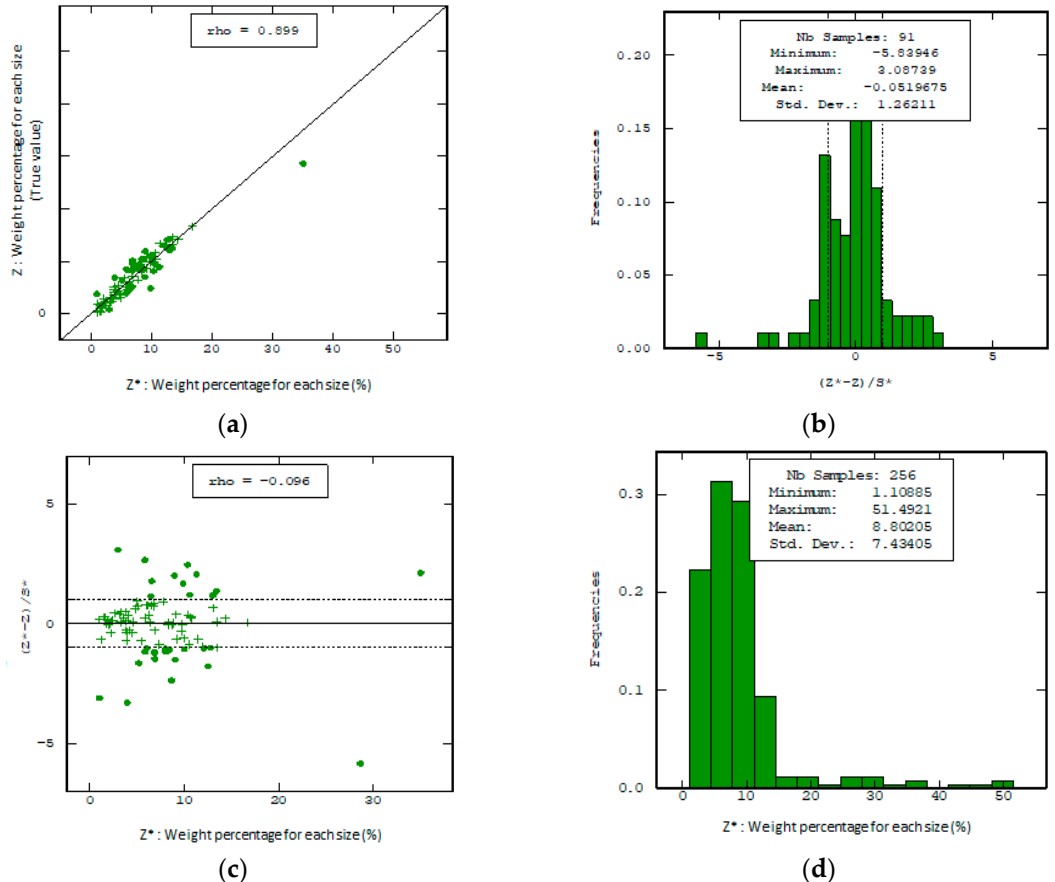

**Figure A5.** Cross-evaluation of the geostatistical analysis of LGP-2 with ore-A. (**a**) Scatter plot (identity line) where the x-axis represents the modeled data, and the y-axis represents the experimental data. (**b**) Standardized error histogram. (**c**) Standardized error based on the estimated value. (**d**) Histogram and its statistical parameters of the estimated data.

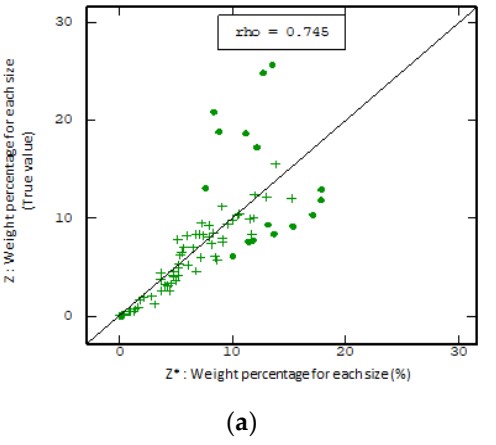

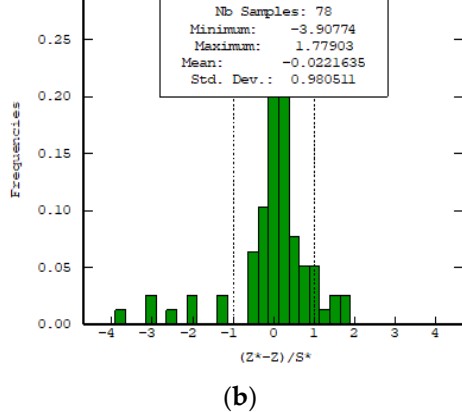

**Figure A6.** *Cont.*

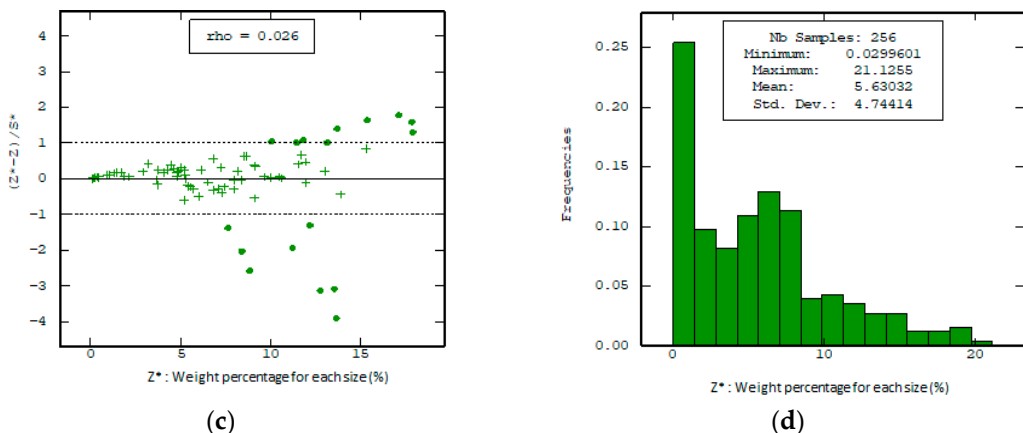

(**c**)　　　　　　　　　　　　　　　　(**d**)

**Figure A6.** Cross-evaluation of the geostatistical analysis of LGP-2 with ore-B. (**a**) Scatter plot (identity line) where the x-axis represents the modeled data, and the y-axis represents the experimental data. (**b**) Standardized error histogram. (**c**) Standardized error based on the estimated value. (**d**) Histogram and its statistical parameters of the estimated data.

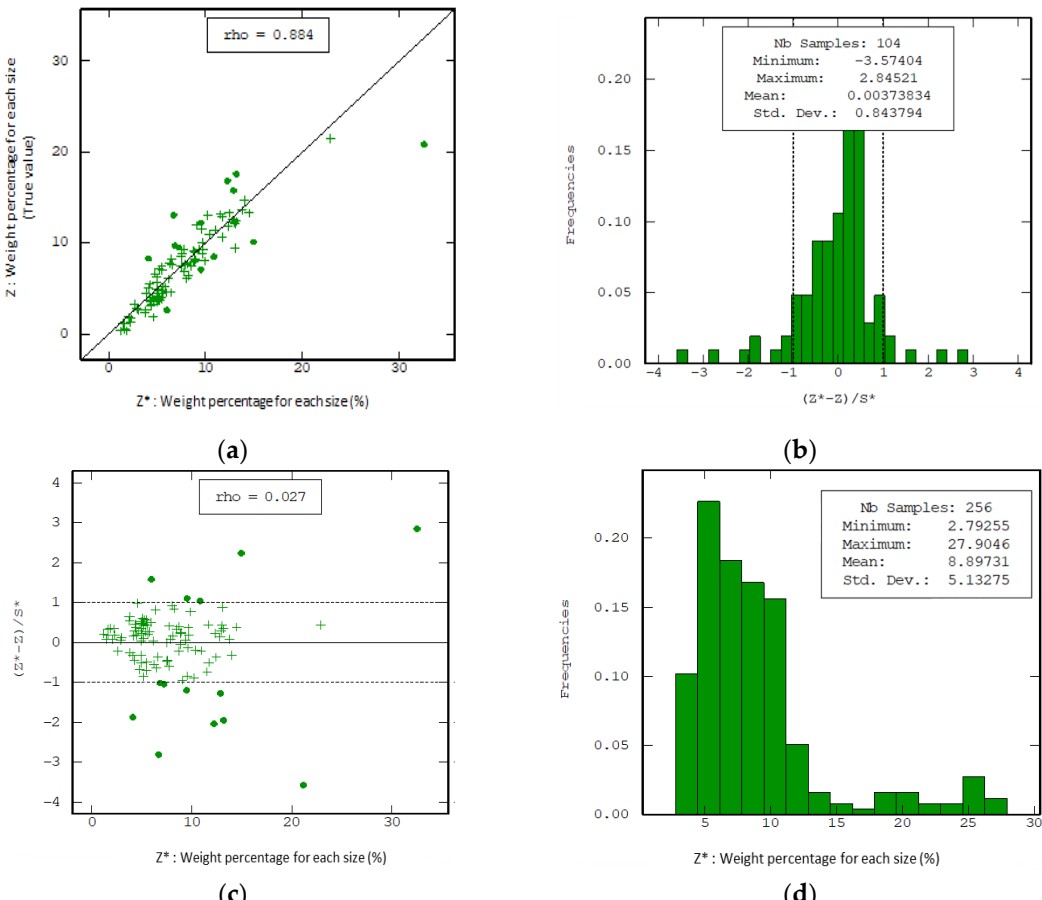

**Figure A7.** Cross-evaluation of the geostatistical analysis of LGP-3 with ore-A. (**a**) Scatter plot (identity line) where the x-axis represents modeled data, and the y-axis represents the experimental data. (**b**) Standardized error histogram. (**c**) Standardized error based on the estimated value. (**d**) Histogram and its statistical parameters of the estimated data.

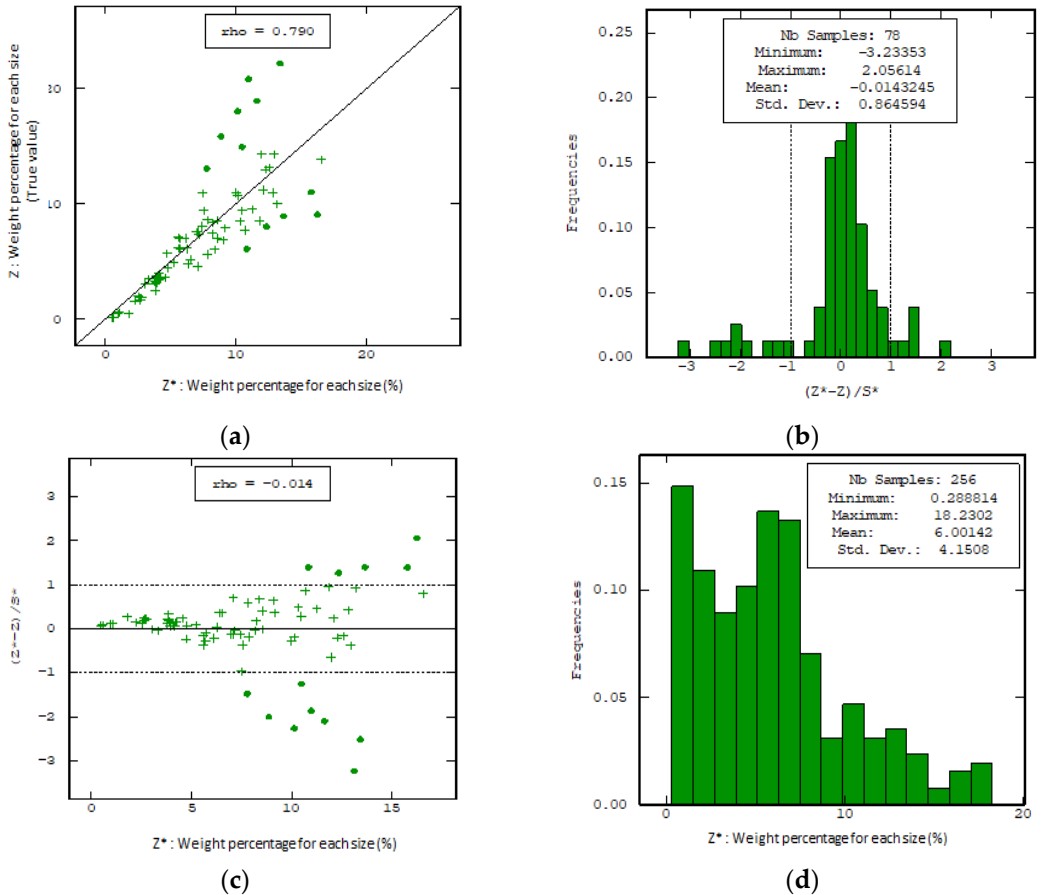

**Figure A8.** Cross-evaluation of the geostatistical analysis of LGP-3 with ore-B. (**a**) Scatter plot (identity line) where the x-axis represents modeled data, and the y-axis represents the experimental data. (**b**) Standardized error histogram. (**c**) Standardized error based on the estimated value. (**d**) Histogram and its statistical parameters of the estimated data.

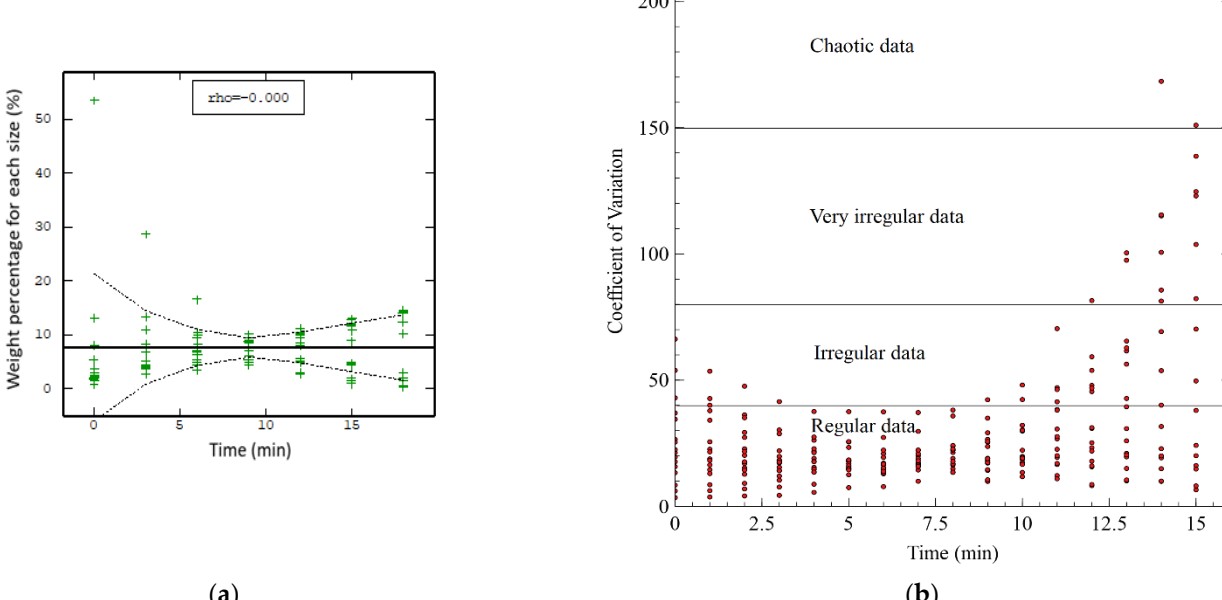

**Figure A9.** *Cont.*

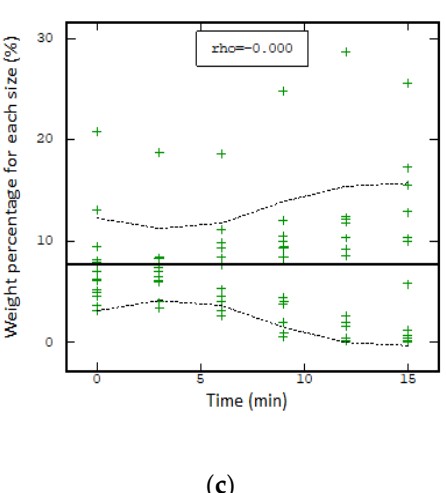

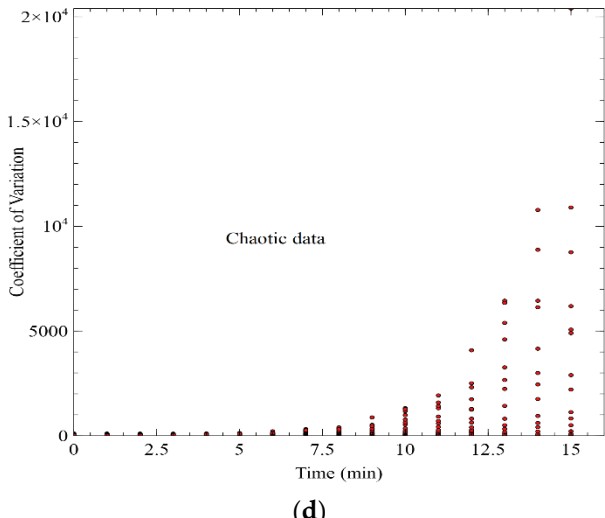

(**c**)

(**d**)

**Figure A9.** Result for LGP-2 for time evolutions. (**a**) Weight of each sizer fluctuation zone of the reconstructed data for ore-A; (**b**) *Cv* for ore-A. (**c**) Weight of each sizer fluctuation zone of the reconstructed data for ore-B; (**d**) *Cv* for ore-b.

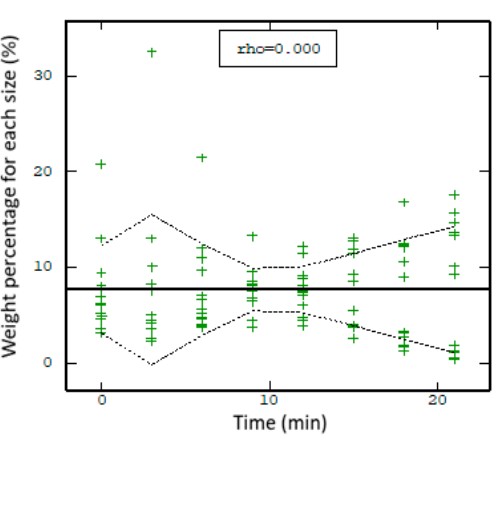

(**a**)

(**b**)

**Figure A10.** *Cont.*

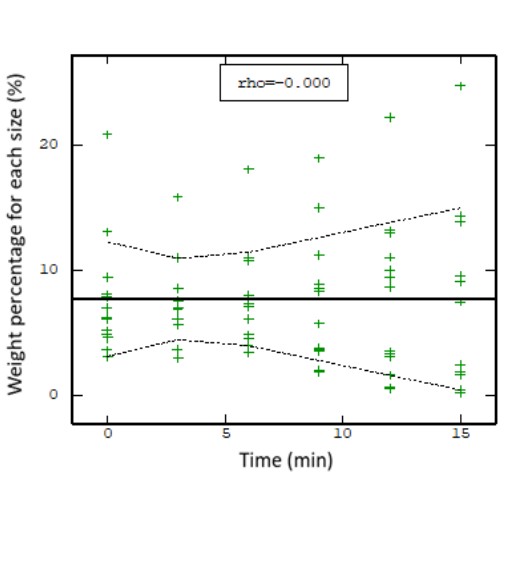

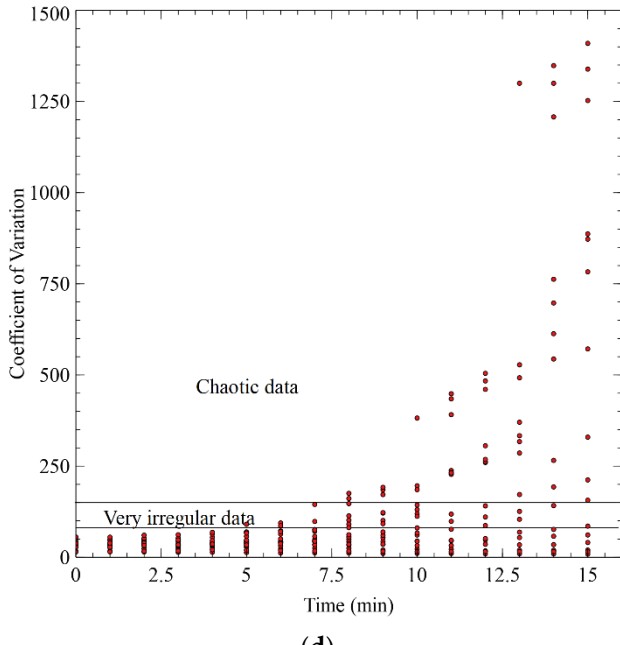

(**c**)                                        (**d**)

**Figure A10.** Results for LGP-3 for the time evolutions. (**a**) Weight of each sizer fluctuation zone of the reconstructed data for ore-A; (**b**) *Cv* for ore-A. (**c**) Weight of each sizer fluctuation zone of the reconstructed data for ore-B; (**d**) *Cv* for ore-b.

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
