# Peer review of "Integration of Lineal Geostatistical Analysis and Computational Intelligence to Evaluate the Batch Grinding Kinetics"

_minerals, doi:10.3390/min12070823_

Round 1

Reviewer 1 Report

The paper ”Integration of lineal geostatistical analysis and computational intelligence to evaluate the batch grinding kinetics” by Dr. Lucay, Dr. Delgado and Dr. Sepúlveda describes a methodology for analyzing and modeling grinding of ore batches. The paper combines theory and practice by describing the methodology and then applying it for two ore batches.

Unfortunately for the review, after reading the full paper, this reviewer has had to realize that the paper is more advanced than my knowledge in the field. While I can give some hopefully useful advice on a number of details, I do not feel qualified to evaluate the overall value of the paper.

MINOR ISSUES AND SUGGESTIONS

Lines 66-84 are a verbatim repetition of lines 49-65.

Line 134: The term “GA” is used without prior definition. Is the meaning “geostatistical analysis” (defined in line 201)? If yes, then please define here. The term “the weight of each GA” is not clear. An analysis may carry weight (e.g. being robust or providing a good fit), but it is not clear from the context how this weight is to be measured.

Line 156: The Introduction ends with both a question mark and a period.

Line 159: “Phase II: mathematical evaluation”. It is suggested to use instead the term “geostatistical evaluation”, as this is the term used elsewhere in the paper, e.g. in line 188.

Line 166: “SPI” is not defined.

Line 254: “ith the restriction”. The meaning appears to be: “with the restriction” (lowercase, as it appears to be a continuation of the previous line; in that case, the line should not be indented).

Lines 295-296: “into a higher-dimensional space, and into w [belongs to] R^n”. The meaning appears to be: “into a higher-dimensional space R^n, w [belongs to] R^n”, i.e. “R^n” in both parts of the sentence, and no “into” in the second part. Correct?

Line 344: It is suggested to divide into two sentences: ... for data analysis”. This comment ...

Lines 374-378: It is suggested to round the weights a bit, e.g. to 10.22 kg, 9.34 kg and 6.66 kg. The last gram is hardly important, and not having three digits after the decimal will help the reader to avoid reading 10.220 kg as 10,220 kg (10 tons), etc.

Lines 432-433: “And were used ... CPU 2.5 GHz 2.7 GHz”. It is suggested to start as “These were used ...” and write only one clock frequency (either 2.5 GHz or 2.7 GHz).

Figure 2: It would be more easy to read the graphs if the colors were presented in the same order as the visual impression (red first, blue last). 

A tip for controlling the graph order in the legend: Excel presents the legends in the order of the input data. The order of the input data can be changed by this procedure:

Click on chart to choose it. Right-click and then “Choose data”. In the dialog that appears, click on a data series, and then click on the up or down arrows to change its order. Continue with other data sets until the preferred order is achieved.

Lines 451-456 and Figure 3b: It is unclear how the adjustment/normalization is performed. Figure 3a and Table A1 indicates that sieve sizes decrease by approximately a factor of sqrt(2) = 1.41 every time. How does this relate to “a constant of 1.8  in the Y-axis”? In Figure 3b, the Y-axis text indicates multiplication by 18 and indicates that the result is still in mm, which does not make sense to this reviewer. (Taking the logarithm of the geometric series would turn it into a linear series, but that does not seem to be what is described or plotted.)

Lines 464-465: The text “accumulation of information on small sample sizes” can be misunderstood, as the phrase “small sample size” is often used about a low number of samples (n low). Suggestion: “the small-size samples”.

Line 484: “Figure 6” is referenced, should this be “Figure 5”? 

Lines 494-495: It is unclear what the numbers in parentheses refer to in the text “the nugget effect (3), lineal scale (5,5,12) and gaussian scale (15,25, sill 20).”

Line 584: A period (or more text) seems to be missing after “to test LS-SVM”.

Line 627: “ore -A” -> “Ore-A”

Line 757: The headline is “Appendix B”, but no other appendix is present. Is the meaning “Appendix A” or perhaps just “Appendix”?

Author Response

Reviewer N°1 (light blue)

MINOR ISSUES AND SUGGESTIONS

Lines 66-84 are a verbatim repetition of lines 49-65.

Thank you for the comment and I apologize for the repetition, the duplicated part has been eliminated.

Line 134: The term “GA” is used without prior definition. Is the meaning “geostatistical analysis” (defined in line 201)? If yes, then please define here.

corrected observation

The term “the weight of each GA” is not clear. An analysis may carry weight (e.g. being robust or providing a good fit), but it is not clear from the context how this weight is to be measured.

Absolutely agree with your comment, it was modified.

Line 156: The Introduction ends with both a question mark and a period.

Eliminated.

Line 159: “Phase II: mathematical evaluation”. It is suggested to use instead the term “geostatistical evaluation”, as this is the term used elsewhere in the paper, e.g. in line 188.

corrected observation, it was modified.

Line 166: “SPI” is not defined.

corrected observation, it was modified.

Line 254: “ith the restriction”. The meaning appears to be: “with the restriction” (lowercase, as it appears to be a continuation of the previous line; in that case, the line should not be indented).

it was modified.

Lines 295-296: “into a higher-dimensional space, and into w [belongs to] R^n”. The meaning appears to be: “into a higher-dimensional space R^n, w [belongs to] R^n”, i.e. “R^n” in both parts of the sentence, and no “into” in the second part. Correct?

You are right; the text is confusing. The word “into” was deleted in the manuscript’s revised version.

Line 344: It is suggested to divide into two sentences: ... for data analysis”. This comment ...

it was modified.

Lines 374-378: It is suggested to round the weights a bit, e.g. to 10.22 kg, 9.34 kg and 6.66 kg. The last gram is hardly important, and not having three digits after the decimal will help the reader to avoid reading 10.220 kg as 10,220 kg (10 tons), etc.

Absolutely agree with your comment, it was modified.

Lines 432-433: “And were used ... CPU 2.5 GHz 2.7 GHz”. It is suggested to start as “These were used ...” and write only one clock frequency (either 2.5 GHz or 2.7 GHz).

Thank you very much for the comment, it was updated and the parameters were modified.

Figure 2: It would be more easy to read the graphs if the colors were presented in the same order as the visual impression (red first, blue last). 

Figure 2 was improved in the manuscript’s revised version.

A tip for controlling the graph order in the legend: Excel presents the legends in the order of the input data. The order of the input data can be changed by this procedure:

Click on chart to choose it. Right-click and then “Choose data”. In the dialog that appears, click on a data series, and then click on the up or down arrows to change its order. Continue with other data sets until the preferred order is achieved.

Dear reviewer, I appreciate the recommendation, but we make the graphs in Rstudio or R, this way we have a better quality of the figure and there is no problem of size changes. in any case, we use the graphs in Excel to make a preliminary graph to have a first approximation, and that case, your observation I will use it, thank you very much.

Lines 451-456 and Figure 3b: It is unclear how the adjustment/normalization is performed. Figure 3a and Table A1 indicates that sieve sizes decrease by approximately a factor of sqrt(2) = 1.41 every time. How does this relate to “a constant of 1.8  in the Y-axis”? In Figure 3b, the Y-axis text indicates multiplication by 18 and indicates that the result is still in mm, which does not make sense to this reviewer. (Taking the logarithm of the geometric series would turn it into a linear series, but that does not seem to be what is described or plotted.)

Lines 464-465: The text “accumulation of information on small sample sizes” can be misunderstood, as the phrase “small sample size” is often used about a low number of samples (n low). Suggestion: “the small-size samples”.

Your observation was correct, there was an error in line 451, therefore, the modification is made from 1.8 to 18, but the units are maintained, because it was an amplification factor of the Y-axis, thus, the mesh becomes sample points with surface coordinates a "little" more regular.

Line 484: “Figure 6” is referenced, should this be “Figure 5”? 

Absolutely agree with your comment, it was modified.

Lines 494-495: It is unclear what the numbers in parentheses refer to in the text “the nugget effect (3), lineal scale (5,5,12) and gaussian scale (15,25, sill 20).”

Se entiende la observación, pero la forma de describir el modelo geostadisticos presentan componentes espaciales, que se definen en el programa isati

Line 584: A period (or more text) seems to be missing after “to test LS-SVM”.

In this remark, there is nothing missing in the acronym.

Line 627: “ore -A” -> “Ore-A”

all were changed to lowercase, except for figures 4, A1 and A2, because before they were with a period followed by a period.

Line 757: The headline is “Appendix B”, but no other appendix is present. Is the meaning “Appendix A” or perhaps just “Appendix”?

it was modified.

Reviewer 2 Report

In the Introduction, paragraphs 66 to 75 are the same as paragraphs 49-57. Paragraphs 78 to 84 are the same as paragraphs 59 to 65.

In section 3.1 of Phase I: Experimental evaluation and traditional quantification, it would be clearer if the baseline data were placed in a table.

This paper has space for further research. As the authors have stated, the kinetic evolution of the milling process can be improved by increasing the number of experimental tests.

Author Response

Reviewer N°2 (Yellow)

In the Introduction, paragraphs 66 to 75 are the same as paragraphs 49-57.

Thank you for the comment and I apologize for the repetition, the duplicated part has been eliminated.

Paragraphs 78 to 84 are the same as paragraphs 59 to 65.

Thank you for the comment and I apologize for the repetition, the duplicated part has been eliminated.

In section 3.1 of Phase I: Experimental evaluation and traditional quantification, it would be clearer if the baseline data were placed in a table.

I considered this option, but I discarded it, because as there are many parameters, I was in doubt as to how many elements to include, and for other hand, including another table makes the document longer than it already is.

This paper has space for further research. As the authors have stated, the kinetic evolution of the milling process can be improved by increasing the number of experimental tests.

Unfortunately, the number of experimental samples is not necessarily sufficient for an adequate evaluation, since it also depends on the uncertainty of the process (this point is evaluated in the publication), rigorous operational protocols and an exhaustive mathematical analysis. finally, there are other mathematical techniques that were not included in this paper, due to the length of the document, but with this, it is possible to propose a more adequate methodology, in order to evaluate in the most correct way, the results of the milling process.